# CCN activity of six pollenkitts and the influence of their surface activity.

Nønne L. Prisle[1,2,3,4], Jack J. Lin[1,3], Sara Purdue[3], Haisheng Lin[4], J. Carson Meredith[4], and Athanasios Nenes[3,4,5,6]

[1]University of Oulu, Nano and Molecular Systems Research Unit, P.O. Box 3000, 90014, University of Oulu, Oulu, Finland
[2]University of Helsinki, Department of Physics, P.O. Box 64, 00014, University of Helsinki, Helsinki, Finland
[3]Georgia Institute of Technology, School of Earth & Atmospheric Sciences, 311 Ferst Drive, Atlanta, GA 30332, USA
[4]Georgia Institute of Technology, School of Chemical & Biomolecular Engineering, 311 Ferst Drive, Atlanta, GA 30332, USA
[5]Institute of Chemical Engineering Sciences (ICE-HT), Foundation for Research, Patras, 26504, Greece
[6]Institute for Environmental Research and Sustainable Development, National Observatory of Athens, 15236, Athens, Greece

**Correspondence:** N. L. Prisle (nonne.prisle@oulu.fi)

**Abstract.** The role of surfactants in governing water interactions of atmospheric aerosols has been a recurring topic in cloud microphysics for more than two decades. Studies of detailed surface thermodynamics are limited by the availability of aerosol samples for experimental analysis and incomplete validation of various proposed Köhler model frameworks for complex mixtures representative of atmospheric aerosol. Pollenkitt is a viscous material that coats grains of pollen and plays important roles in pollen dispersion and plant reproduction. Previous work suggests that it may also be an important contributor to pollen water uptake and CCN activity. The chemical composition of pollenkitt varies between species but has been found to comprise complex organic mixtures including oxygenated, lipid, and aliphatic functionalities. This mix of functionalities suggests that pollenkitt may display aqueous surface activity, which could significantly impact pollen interactions with atmospheric water. Here, we study the surface activity of pollenkitt from six different species and its influence on pollenkitt hygroscopicity. We measure cloud droplet activation and concentration dependent surface tension of pollenkitt and its mixtures with ammonium sulfate salt. Experiments are compared to predictions from several thermodynamic models, taking aqueous surface tension reduction and surfactant surface partitioning into account in various ways. We find a clear reduction of surface tension by pollenkitt in aqueous solution and evidence for impact of both surface tension and surface partitioning mechanisms on cloud droplet activation potential and hygroscopicity of pollenkitt particles. In addition, we find indication of complex non-ideal solution effects in a systematic and consistent dependency of pollenkitt hygroscopicity on particle size. The impact of pollenkitt surface activity on cloud microphysics is different from what is observed in previous work for simple atmospheric surfactants and more resembles recent observations for complex primary and secondary organic aerosol, adding new insight to our understanding of the multifaceted role of surfactants in governing aerosol–water interactions. We illustrate how the explicit characterization of pollenkitt contributions provides the basis for modeling water uptake and cloud formation of pollen and their fragments over a wide range of atmospheric conditions.

## 1 Introduction

Surface tension depression from organic surfactants has long been recognized to impact cloud droplet forming potential of organic aerosol (OA) predicted from equilibrium Köhler theory (Köhler, 1936). Early work by Shulman et al. (1996) and Facchini et al. (1999, 2000) neglected the mass balance of surfactants assumed to be contained in the aerosol as result of their bulk-to-surface partitioning. Later work recognized this limitation (Sorjamaa et al., 2004) and subsequent studies for model atmospheric aerosol systems comprising e.g. fatty acids (Prisle et al., 2008, 2010b; Forestieri et al., 2018), model-HULIS (Kristensen et al., 2014), organo-sulfates (Hansen et al., 2015), and model-generated synthetic aerosol representing a variety of atmospheric environments (Lowe et al., 2016), demonstrated how surface activity and its effect on cloud condensation nuclei (CCN) activity involve complex non-linear interactions between both surface tension and bulk-to-surface partitioning in droplets. As a result, the overall impact of surfactants on atmospheric cloud droplet formation is difficult to quantify in a straightforward manner and remains to be firmly established (Prisle et al., 2012; Lowe et al., 2016). Two factors must be highlighted in this context. First, it is still not possible to directly measure the surface tension of dilute aqueous submicron activating droplets containing surfactants, and all modeling is thus based on downscaling macroscopic, or bulk phase, measurements to droplet size systems using equilibrium thermodynamic relations (e.g. Prisle et al., 2008, 2010b; Topping, 2010; Raatikainen and Laaksonen, 2011). Measurements that have been able to directly quantify droplet interfacial tensions using optical tweezers (Bzdek et al., 2016) or biphasic microfluidic devices (Boyer and Dutcher, 2017) have relied on optical techniques that require studying droplets larger than the typical size of droplets at the activation threshold. Second, detailed characterization of the impact of surface activity on droplet activation thermodynamics has to date only been made for a small ensemble of OA systems, with limited breadth of chemical functionality and complexity (Petters and Petters, 2016). Gas-phase compounds may also act as surfactants (Romakkaniemi et al., 2011; Sareen et al., 2013), further complicating the potential interactions that occur in atmospherically relevant systems.

Various approaches have been used to account for effects of organic surface activity in Köhler model frameworks. Padró et al. (2007) proposed using so-called Köhler Theory Analysis (KTA) derived from simple Köhler theory to infer average values of surface tension and bulk molar volume of an aerosol system with known and systematic perturbations on the composition, usually by adding a known amount of electrolyte to an unknown organic mixture. Other work focused on more comprehensive approaches for thermodynamically consistent predictions of these properties (e.g. Sorjamaa et al., 2004; Prisle et al., 2008, 2010b), and simplified schemes including various approximations to enhance computational efficiency (e.g. Topping, 2010; Raatikainen and Laaksonen, 2011). Based on results from the comprehensive models, Prisle et al. (2011) proposed a simple representation of the overall CCN activity of surface active organic aerosol to facilitate a computationally efficient approach to study larger-scale effects. The simple approach essentially assumes an effective hygroscopicity value (Petters and Kreidenweis, 2007) of zero for surfactants. In global predictions of cloud droplet numbers and cloud radiative forcing, the resulting effect of surface activity was therefore found to be negligible (Prisle et al., 2012).

Recent work however suggests that there may indeed be a suite of aerosol systems for which the impact of droplet surface tension at the point of activation could be more significant than previously indicated. Ruehl et al. (2016) demonstrated with a framework based on very similar assumptions to those of Prisle et al. (2011) that surface tension can significantly enhance droplet hygroscopicity and CCN activation of chamber generated secondary organic aerosol (SOA). Ovadnevaite et al. (2017) soon after showed that there may be evidence for such mechanisms in observations of primary organic aerosol (POA) from a coastal environment at Mace Head, with potential global implications due to the relative significance of marine aerosol. However, our understanding of the impact of surface active OA on cloud microphysics and atmospheric predictions of CCN and cloud droplet number concentrations is still incomplete. Specifically, the various existing Köhler-based model frameworks for evaluating aerosol–water interactions and cloud forming potential remain to be validated for a much broader ensemble of atmospheric OA compound classes, chemical functionality, and complexity. Here, we study real atmospheric samples of water soluble and surface active biogenic POA with a complex mix of chemical functionalities, extracted from pollen grains. To our knowledge, such mixtures have not previously been subject to detailed analysis for effects of surface activity on cloud microphysical properties.

Pollen is the male gametophyte of seed plants and plays a critical role in plant reproduction (Punt et al., 2007). Pollen grains can disperse in air and adhere on insects or other animals allowing for their distribution over large areas (Bedinger, 1992; Pacini, 2010). Pollen possesses a range of ornamentations that vary in morphology (i.e., spherical, granulate, reticulate, and echinate) and feature size, with nominal particle diameters of 3–250 $\mu$m. Pollen of entomophilous and zoophilous species are coated by a viscous material that enables numerous functions, including adhesion and transport by pollinators (Pacini and Hesse, 2005). This substance, typically called *pollenkitt*, is formed from the final degradation of the plant tapetum during pollen development, resulting in the deposition of a coating rich in plastid-derived lipids and other pigmented compounds (Pacini, 1997; Dickinson et al., 2000; Knoll, 1930). Pollen transmission is facilitated by the ability of pollenkitt to adhere pollen grains to one another and to pollinators during transport.

While the chemical composition of pollenkitt has been characterized in a number of studies (Piffanelli et al., 1998), relatively little is known about its physico-chemical properties. Pollenkitt is comprised of a mixture of saturated and unsaturated lipids, carotenoids, flavonoids, proteins, and carbohydrates (Piffanelli et al., 1998; Pacini and Hesse, 2005; Dobson, 1988). Analysis of the lower molecular weight compounds from the pollenkitt of *Rosa rugosa* revealed aromatics, C11-C16 aliphatics, terpenoids, and C16 acetate, with few of these compounds also found in *Rosa canina* (Dobson et al., 1987). From a collection of pollen from 69 angiosperms, Dobson (1988) found the pollenkitt to contain non-glyceride neutral lipids (hydrocarbons, fatty acid methyl esters, sterol esters, aldehydes, and ketones), a few polar lipids, and pigments (yellow carotenoids and flavonoids).

Interestingly, in pollen from some plant species the pollenkitt serves as a barrier to reduce the rate of water loss, preserving the viability of the pollen, which decreases with dehydration (Pacini and Hesse, 2005). The polar components in pollenkitt should support some absorption of water, and many of these substances are expected to be water soluble themselves. Previous quantitative studies of pollen adhesion to varied surfaces by atomic force microscopy (Lin et al., 2013) showed that pollenkitt significantly enhances pollen adhesion and that the effect is driven by the formation of pollenkitt capillary bridges. A subsequent study found that pollenkitt hydration by water uptake at high humidity changed pollenkitt properties and its capillary

adhesion (Lin et al., 2015). This finding has important implications for understanding pollenkitt's role in plant reproduction, allergy and asthma, as well as the role of pollen as atmospheric condensation nuclei.

The hygroscopic water uptake of whole pollen grains was studied at subsaturated conditions by Pope (2010) and Griffiths et al. (2012) who found that pollen of seven different species were moderately hygroscopic with water-wettable surfaces. Using an electrodynamic balance, Pope (2010) determined values for the hygroscopicity parameter $\kappa$ to be between 0.05 and 0.1 for four different types of pollens (daffodil, water birch, pussy willow, and black walnut), while Griffiths et al. (2012) found $\kappa$ values between 0.08 and 0.17 for six different pollens (rye, firebush, giant sage, daffodil, water birch, and pussy willow.) Due to their large grain sizes, they concluded that pollen grains would be effective CCN even at low supersaturations. However, suspended number concentrations of whole pollen grains are often too low to expect significant impact on cloud microphysics. Steiner et al. (2015) characterized the cloud droplet forming potential (CCN activity) of so-called submicron subpollen particles (SSP), which form as fragments from whole pollen grains. Laboratory experiments found whole pollen grains can rupture and release SSP when wetted by direct contact with liquid water or exposure to high ambient relative humidities of 80–96% (Grote et al., 2001; Taylor et al., 2002; Grote et al., 2003; Taylor et al., 2004). Steiner et al. (2015) found such SSP of six pollen types characteristic of mid-latitudes to be active enough to serve as CCN, with particles of 200–250 nm diameters activating at atmospheric conditions below 1% supersaturation. From these and the above mentioned studies, it is evident that pollen take up atmospheric water and that pollenkitt plays a role in governing the various interactions involved. Nevertheless, how much of the observed hygroscopicity of pollen is driven by water adsorption onto the insoluble surfaces vs. absorptive water uptake by the pollenkitt coatings is currently unclear.

Here, we therefore focus on understanding the water-uptake properties of the pollenkitt mixture specifically. Depending on the thickness of its pollenkitt coating, the water uptake of a pollen particle depends on the ability of the pollenkitt mixture to depress the effective water activity through solute and surface tension effects. Based on the current knowledge of pollenkitt chemical composition, comprising a suite of organic compounds with both hydrophilic and hydrophobic moieties, we might expect the pollenkitt mixture to exhibit some degree of both solubility and surface activity in aqueous solution, analogous to e.g. SOA (McNeill et al., 2014). We obtain pollenkitt samples from pollen of six different plant species and study their specific potential to reduce aqueous surface tension and promote water uptake to the aerosol phase for a wide range of particle sizes and supersaturated humidities. The analysis of CCN activity at water supersaturation conditions enable us to highlight aqueous phase mixing effects which may be masked by higher aqueous concentrations in subsaturated hygroscopic growth measurements and thus access an aerosol phase regime with anticipated concurrent bulk-to-partitioning and surface tension effects, which may then be interpreted in a detailed thermodynamic framework via Köhler theory. The characterization of hygroscopic water uptake and cloud activation generally has implications beyond cloud microphysics, for all phenomena governed by the same underlying aerosol-water interactions. Specifically for pollenkitt as OA component, these include, but are not limited to, the atmospheric growth, transport, ageing, and cloud formation of SSP and pollen as giant CCN and IN, and biological functions related e.g. to pollen wettability and adhesive properties.

## 2 Methods

### 2.1 Pollenkitt sample preparation

Pollenkitts were obtained from the pollen grains of six different species, including olive (*Olea europaea*), black poplar (*Populus nigra*), Kentucky bluegrass (*Poa pratensis*), common ragweed (*Ambrosia artemisiifolia*), common dandelion (*Taraxacum officinale*), and common sunflower (*Helianthus annuus*), as previously studied at subsaturated conditions (except Kentucky bluegrass) by Lin et al. (2013). All the native non-defatted pollen grains were purchased from Greer Laboratories (Lenoir, NC), stored at 0 °C, and were used as received without further purification. In the case of the dandelion pollen, the pollen grains were obtained off of honey bees instead of the flowers directly so that the dandelion sample is expected to include some amount of nectar (sugars dissolved in water) from the honey bees (Sladen, 1912; Campos et al., 2008). The extraction of pollenkitt from each native pollen grain was performed by adding a known mass of each pollen sample to 40 mL deionized (DI) water and shaking by a rotational shaker (Barnstead Thermolyne Labquake Shaker Rotisserie) for 10–15 min. Then the solutions were centrifuged (2.7 × 1000 rpm, 20 min) in an IEC Centra CL2 centrifuge (Thermo Scientific) and subsequently filtered to remove all the pollen particles. The resulting solutions were diluted with DI water to concentrations suitable for the appropriate CCN and surface tension measurements. Solutions were prepared such that the extracted amount of pollenkitt was completely dissolved in the aqueous solution and no phase separation was observed. Surface tension measurements were made over a pollenkitt concentration range of 0.01–200 mg $L^{-1}$. CCN measurements were made at pollenkitt concentration of 0.05 mg $L^{-1}$, except for olive pollenkitt which was made at 0.5 mg $L^{-1}$. This method of extracting atmospheric POA from commercially available pollen grains allowed us to obtain sufficient amounts of sample to perform both CCN and surface tension analysis in parallel. Limited sample availability is one of the key challenges determining the scarcity of similar studies so far made for atmospheric OA.

Aqueous solutions of pollenkitt along with ammonium sulfate (AS) were also prepared with pollenkitt-to-AS mass ratio of 4:1, so that AS comprises 20% of the total solute mass. Ammonium sulfate was selected as a well-characterized proxy for the inorganic fraction of atmospheric aerosol, and we focused on poplar and ragweed pollenkitt as the most and least CCN active of the six pollenkitts investigated (see Section 3.1 below). Mixing with ammonium sulfate salt in the aqueous phase is a simple way to mimic atmospheric aging in various environments, such as cloud processing and formation of secondary inorganic aerosol in polluted air, but is also used here as a physico-chemical indicator to highlight the presence and magnitude of characteristic signatures of aqueous surface activity (Prisle et al., 2010b, 2011). The specific organic–inorganic mass mixing ratio was chosen based on observations from previous work that (*i*) surface activity effects became evident in cloud droplet activation behavior of particles with more than about 50% by mass of surface active organic aerosol (Prisle et al., 2008, 2010b), (*ii*) additional effects of organic–inorganic solute interactions were predicted to be most prominent for mass mixing ratios in the range of 80-95% surface active organic mass (Prisle et al., 2010b, 2011), and (*iii*) among these particle compositions, the lower ratios of surface active organic are likely to be the more atmospherically relevant in general (McFiggans et al., 2006; Hallquist et al., 2009). However, as pollenkitt is a pollen grain borne POA, the actual range of organic–inorganic mixing ratios resulting from atmospheric processing remain speculative. A full characterization of mixing effects with the methods applied

here would require preparation of fresh stock solutions for each organic–inorganic composition. Due to the relative scarcity of pollenkitt samples, measurements were therefore here limited to one AS mixing ratio for each of the two pollenkitt mixtures.

## 2.2 CCN experiments

Size-resolved CCN activity of pollenkitt and their mixtures with ammonium sulfate was measured using Scanning Mobility CCN Analysis (SMCA; Moore et al., 2010). A schematic and flow diagram of the experimental setup for CCN activation experiments closely resembling that used here is presented by (Padró et al., 2007). At the time of each experiment, prepared 50 mL aqueous samples were transferred to a large volume nebulizer (Hudson RIC, Ref. 1770). Filtered laboratory air was pumped into the nebulizer at 5 psig with the nebulizer set to a flow rate of 10 L min$^{-1}$ and 98% $O_2$. The generated particles were dried to roughly 10% RH in two diffusion driers and charged neutralized (TSI 3077) before being sent to a differential mobility analyzer (DMA, TSI 3080). The DMA voltage was continuously cycled to sample particles sizes in the range of 7–512 nm over 135 s. The size-classified aerosol was split between a condensation particle counter (TSI CPC 3010) to measure total particle number and a cloud condensation nuclei counter (Droplet Measurement Technologies CCN-100) to measure activated particle number as a function of supersaturation. An inversion procedure was used to calculate the fraction of total particles at a given size that have activated into droplets (Moore et al., 2010). For each DMA size distribution, these data were fit to a sigmoidal function after correcting for diffusional losses and multiply-charged particles in the DMA. The particle dry diameter at which 50% of the particles activate into droplets, $D_{p,50}$, corresponds to a particle with critical supersaturation equal to the instrument supersaturation. This is here referred to as the critical dry diameter, following the terminology of e.g. Rose et al. (2010) and Kristensen et al. (2014). The CCN counter was operated at nine different supersaturations (0.10, 0.21, 0.38, 0.51, 0.69, 0.90, 1.0, 1.3, and 1.4%) for 20 minutes each so that approximately eight complete size distributions from the DMA are sampled at each supersaturation. Particle number concentrations introduced to the CCN counter were less than 1000 cm$^{-3}$ to avoid water vapor depletion effects in the CCN counter (Lathem and Nenes, 2011).

In addition to providing specific characterization of the cloud forming potential of particles with various compositions, in a wider sense CCN activity analysis is also a means to obtain information of water interactions thermodynamics for very small amounts of sample material. As particles of different sizes can be produced from a single stock solution to access all supersaturations scanned by the instrument, our analysis here focuses on characterization with variation of humidity and particle size, rather than detailed resolution of aerosol component mixing effects.

## 2.3 Surface tension measurements

In order to investigate the specific impact of surface tension on pollenkitt CCN activation, we measured concentration-dependent surface tension of poplar and ragweed pollenkitt in their pure aqueous solutions and in aqueous solution with ammonium sulfate. Surface tension of binary and ternary aqueous pollenkitt solutions were measured via axisymmetric drop shape analysis of pendant drops in air with a ramé-hart goniometer (Model 250). Briefly, a pendant drop of solution was created by using a syringe with a steel needle, and a charge-coupled device (CCD) camera captured the variation of drop shape. The surface tension was obtained by analyzing the contour shape resulting from the balance of gravitational and surface forces

through solution of the Young-Laplace equation as described elsewhere (Berry et al., 2015). All experiments were performed at room temperature of 21 °C.

Samples for surface tension measurements were prepared in the same way as for CCN measurements described above. Concentrations ranged from 0.01–200 mg pollenkitt per L solution, all of which were observed during sample preparations to be within the aqueous solubility range of each pollenkitt mixture.

## 2.4 Köhler modeling

Cloud droplet activation of pollenkitt particles was modeled using different models previously developed by Prisle and co-workers. All models are based on equilibrium Köhler theory in the form

$$S \equiv \frac{p_w}{p_w^0} = a_w \exp\left(\frac{4\nu_w \sigma}{RTd}\right) \tag{1}$$

where $p_w$ is the equilibrium water partial pressure over an aqueous droplet, $p_w^0$ is the saturation vapor pressure of pure water at the temperature $T$ (in Kelvin) in question, $S$ is equilibrium water vapor saturation ratio for the stable droplet, $a_w$ is the water activity of the droplet solution, $\nu_w$ is the partial molar volume of water in this solution, $\sigma$ is the droplet surface tension, $R$ is the universal gas constant, and $d$ is the spherical solution droplet diameter. For growing droplets, the equilibrium Köhler curve and threshold for droplet CCN activation, the critical saturation ratio ($S_c$) and corresponding critical droplet diameter ($d_c$), are iterated for each dry particle size and composition (pollenkitt, PK, or pollenkitt mixtures with ammonium sulfate, PK–AS). Droplets that have been exposed to ambient water saturation ratios larger than their respective threshold values ($S \geq S_c$), and thus surpassed their critical droplet diameters ($d \geq d_c$), are assumed to be activated cloud droplets.

Common for all models, calculations are initiated by determining the total amount of pollenkitt and ammonium sulfate solute in the aqueous droplet phase, $c_{\text{PK}}^T$ and $c_{\text{AS}}^T$. The initially dry particles are assumed to be spherical with volume-equivalent diameters corresponding to the electrical-mobility diameter mode selected by the DMA during experiments, $D_p = D_p^e$ (Sorensen, 2011). Although pure AS particles have been observed to have shape factors of 1.04, we here assume that the fraction of organic component is too high to support AS crystal structure and particles are assumed to be spherical. The dry particle volume, which determines the amount of solute in the growing droplets, is thus obtained as $V_p = \frac{\pi}{6}D_p^3$. We then assume volume additivity of the various chemical components $i$ in the dry particles to obtain the dry particle mass density as

$$\rho_p = \left(\sum_i \frac{W_{p,i}}{\rho_i}\right)^{-1}, \tag{2}$$

where $W_{p,i}$ is the dry particle mass fraction and $\rho_i$ the bulk mass density of each individual component $i$, respectively. With the dry particle mass $\rho_p V_p$, the mass of each component can be found from the corresponding mass fraction and given the molecular weights $M_i$, also the molar amounts of all particle components can be determined.

The density of ammonium sulfate is well-known as $\rho_{\text{AS}} = 1.769$ g cm$^{-3}$. For organic aerosol mixtures, a range of densities has been measured in the atmosphere, often between 1.2 and 1.7 g cm$^{-3}$ (Hallquist et al., 2009), but significantly lower values have also been found (Kannosto et al., 2008). The individual components of OA and their respective bulk material properties,

including mass densities, often remain unknown and one approach has been to assume unit density for these compounds and their mixtures ($\rho_{OA} = 1$ g cm$^{-3}$) while awaiting future findings (Prisle et al., 2010a). The pollenkitts investigated here are all complex mixtures of a suite of different organic compounds. For the Köhler model calculations, we therefore initially varied the average effective mass densities of each pollenkitt organic mixture between 0.8 and 1.2 g cm$^{-3}$. In all cases, the higher mass densities give better agreement with measured CCN activities, and only results for those conditions are shown here. We then used KTA (Padró et al., 2007) to estimate the average effective molecular weight of each pollenkitt mixture, consistent with a mass density of 1.2 g cm$^{-3}$ and a range of approximate surface tensions, as will be discussed further in the Results section below.

We here use four different Köhler models, each based on a different set of assumptions regarding droplet surface tension and bulk-to-surface partitioning of the surface active species in aqueous solution. The three thermodynamically consistent models developed in previous work (Prisle et al., 2008, 2010b) and described in detail therein are:

1. The most comprehensive full partitioning model, which explicitly iterates the bulk-to-surface partitioning of surface active species at every aqueous droplet size and dilution state and evaluates droplet surface tension and water activity based on the resulting bulk solute concentrations $c_i^B$ and surface excess of the partitioning equilibrium: $\sigma = \sigma(c_i^B)$ and $a_w = a_w(c_i^B)$, where bulk and total concentrations for surface active species, here pollenkitt, are generally different, $c_{PK}^B \neq c_{PK}^T$.

2. The bulk aqueous solution model, where depletion effects from bulk-to-surface partitioning of surface active species on the bulk composition are neglected, and droplet surface tension and water activity are evaluated based on the total solute concentrations in aqueous solution, i.e. as for the partitioning model, but assuming $c_i^B = c_i^T$ for all solute species.

3. The constant surface tension of pure water model, where surface activity of any droplet component is neglected, such that the droplet bulk concentrations of all solutes is assumed to be directly given by the total droplet composition, and reduction of aqueous surface tension from that of pure water is neglected, i.e. as for the bulk model $c_i^B = c_i^T$ for all solute species and in addition $\sigma = \sigma_w$.

In each of these models, the impact of solute specific interactions of pollenkitt and ammonium sulfate on water activity is neglected, i.e. $a_w \equiv \gamma_w x_w \approx x_w$, where $x_w$ is the mole fraction concentration of water in solution. This is done due to lack of information of the respective activity coefficients and their concentration dependence in solution, but as we discuss below, it may not always be a fully sufficient assumption.

We also used the simple partitioning scheme introduced by Prisle et al. (2011), where the bulk-to-surface partitioning of surface active species is essentially taken to the extreme, such that the droplet bulk is entirely depleted of surfactant species, with resulting concentration $c_{PK}^B = 0$. This implies that both the effective hygroscopicity parameter (see Section 2.6) of pollenkitt in the droplets and the droplet surface tension reduction from the pure water value vanishes, i.e. $\kappa_{PK} = 0$ and $\sigma = \sigma(c_{PK}^B = 0) = \sigma_w$ as described by Eq. (3) when neglecting the secondary contribution from the binary salt (see Section 2.5). The resulting droplet bulk solution consists of binary aqueous ammonium sulfate. The impact of AS dissociation and its ions on water activity

**Table 1.** Main assumptions regarding surface tension ($\sigma$), bulk-to-surface (B–S) partitioning, and water activity, in terms of the mole fraction based water activity ($a_w = \gamma_w x_w$) coefficient $\gamma_w$, in the four Köhler models used.

| Model | B–S partitioning | $\sigma$ reduction | $\gamma_w$ |
|---|---|---|---|
| partitioning | yes, conc. dep. | yes | 1 |
| bulk | no | yes | 1 |
| water | no | no | 1 |
| simple | yes, complete | no | $\frac{1}{x_w}\left(1 - 0.039767 b_{AS} + 0.0079808 b_{AS}^2 - 0.0022641 b_{AS}^3\right)$ (Prisle, 2006) |

is accounted for using a parametrization (Prisle, 2006) based on data from Low (1969). The simple partitioning model was previously shown to work well for predicting the impact of strong surfactants on CCN activity of mixed surfactant–salt model particles comprising sodium dodecyl sulfate (SDS) and the atmospheric fatty acid salts decanoate and dodecanoate. However, it works somewhat less well for sodium octanoate, which is comparatively less surface active (Prisle et al., 2011).

5    An overview of the assumptions distinguishing the four Köhler models is given in Table 1. A schematic of the four Köhler models is furthermore given in Fig. 1 of Prisle et al. (2012).

## 2.5  Surface tension parametrization

In model calculations where the impact of pollenkitt on droplet surface tension is taken explicitly into account, surface tension is evaluated from a parametrization based on the Szyskowski-Langmuir equation (Szyskowski, 1908) in the form

$$\sigma = \sigma_w + \left(\frac{d\sigma_{AS}}{dc_{AS}}\right) c_{AS} - a \ln\left(1 + \frac{c_{PK}}{b}\right). \tag{3}$$

Here, $c_{AS}$ and $c_{PK}$ are the (bulk, superscript omitted, since in macroscopic solution, $c_i^B = c_i^T$) mass concentrations (g solute per L of solution) of ammonium sulfate and pollenkitt, respectively, and $\sigma_w = 72.2$ mN m$^{-1}$ is the surface tension of pure water at 296.65 K. The measured surface tensions for bulk aqueous solutions of each pollenkitt were fitted to Eq. (3). The surface tension gradient for binary aqueous AS, $\left(\frac{d\sigma_{AS}}{dc_{AS}}\right) = 0.01655$ mN m$^{-1}$/g L$^{-1}$ was obtained by linear fitting to the data from Low (1969) and fitting parameters $a$ and $b$ in Eq. (3) have dimensions of mN m$^{-1}$ and g L$^{-1}$, respectively.

## 2.6  $\kappa$ values

The relationship between particle dry diameter and CCN activity can be parameterized using a single hygroscopicity parameter (Petters and Kreidenweis, 2007). The parameter, $\kappa$, is defined based on standard Köhler theory in the form

$$S(d) = \frac{d^3 - D_p^3}{d^3 - D_p^3(1-\kappa)}\exp\left(\frac{4\sigma M_w}{RT\rho_w d}\right), \tag{4}$$

**Table 2.** Pollenkitt CCN activity: Slope of the $\ln D_{p,50}$ vs. $\ln SS$ plot and average hygroscopicity value $\kappa$ for each particle composition.

| Sample | $\ln - \ln$ slope | $\chi^2$ | $\kappa$ |
|---|---|---|---|
| Dandelion | -1.50±0.0299 | 0.01729 | 0.19±0.0162 |
| Kentucky bluegrass | -1.49±0.0237 | 0.01105 | 0.21±0.0162 |
| Olive | -1.44±0.0298 | 0.01873 | 0.17±0.0219 |
| Poplar | -1.41±0.0308 | 0.02094 | 0.24±0.0334 |
| Ragweed | -1.44±0.0355 | 0.02665 | 0.14±0.0203 |
| Sunflower | -1.48±0.0339 | 0.02309 | 0.15±0.0159 |
| Poplar + 20 mass% AS | -1.85±0.0900 | 0.07368 | 0.22±0.0739 |
| Ragweed + 20 mass% AS | -1.63±0.0133 | 0.1920 | 0.28±0.0978 |

where, analogous to Eq. (1), $S(d)$ is the saturation ratio over an aqueous solution drop with size $d$, $D_p$ is particle dry diameter, $\sigma$ is the droplet surface tension, $M_w$ is the molecular weight of water, $R$ is the universal gas constant, and $\rho_w$ is the density of water.

Pairs of particle dry diameter $D_p$ and critical supersaturation $SS_c$ values from SMCA experimental data and modeling data

are used to calculate $\kappa$ from Eq. (4). In the case of experimental data, $D_p$ is represented by the critical dry diameter $D_{p,50}$ and $SS_c$ by the corresponding instrument $SS$. Both $\kappa$ and $d$ are varied independently in order to minimize the difference between the theoretical and measured supersaturation (Rose et al., 2010). This was done because the simplified relations provided by Petters and Kreidenweis (2007) are only valid for $\kappa > 0.2$, a condition which is often not met by our measurements, as seen in Section 3. Furthermore, we have assumed, for the purposes of this calculation, that $\sigma = \sigma_w$ such that any effects of changes

in droplet surface tension on cloud droplet activation are captured by the evaluated effective $\kappa$ parameter. For measurements of each pollenkitt and pollenkitt-ammonium sulfate mixture, averages and standard deviations in $\kappa$ are calculated for each supersaturation and across all supersaturations.

## 3   Results

### 3.1   Pollenkitt CCN activity

Measured CCN activity, given as particle critical dry diameter ($D_{p,50}$) vs. supersaturation ($SS$), for the six different pollenkitts and the two mixtures with ammonium sulfate is shown in Fig. 1. All the pure pollenkitts exhibit similar CCN activity, with some moderate variations between different species. Overall, the CCN activity of the six pollenkitts increase in the order ragweed $\approx$ sunflower $<$ olive $<$ dandelion $<$ Kentucky bluegrass $<$ poplar at a supersaturation of 1.0%, and ragweed $<$ sunflower $<$ olive $<$ dandelion $<$ poplar $<$ Kentucky bluegrass at a supersaturation of 0.2%. See panel (a) in Fig. 1. Hence, the order of

increasing CCN activity varies only little with supersaturation and thus particle size.

For each particle composition, linear fits have been made to the plots of $\ln D_{p,50}$ vs. $\ln SS$. According to standard equilibrium Köhler theory, the slope of these lines should ideally be $-3/2$, and any deviations from this value would indicate the presence of size-dependent variation in particle CCN activation (albeit not which underlying size-dependent property is responsible for the variation). In cases such as this, where the well-constrained laboratory generated particle composition can be assumed to be the same for particles of all sizes generated from the same stock solution, a size-dependent variation in CCN activity could result from either aqueous solubility effects in the droplet bulk or from surface tension effects pertaining to the droplet surface.

Fitted slope values with standard deviations and goodness of fit, $\chi^2 = \sum_i (O_i - E_i)^2$ are given in Table 2. The goodness of fit is the sum of differences between observed outcomes $O_i$ and expected outcomes $E_i$. For dandelion, Kentucky bluegrass, and sunflower, we see no significant deviation from a slope of $-3/2$, and thus no immediate indication in our data of size-dependent variations in CCN activity of these pure pollenkitts. For olive, poplar, and ragweed, the slopes deviate from $-3/2$ beyond the standard deviation of the fit. If size-dependent CCN activity effects are introduced by pollenkitt surface activity, we would therefore expect to find them specifically for these pollenkitts among our samples. Of these, the deviation is strongest for poplar, which is also the most CCN active pollenkitt over most of the particle size range studied. Ragweed is the least CCN active of the six pollenkitts, and has a slope similar to that of olive. We therefore chose to study possible effects of surface tension closer for the cases of ragweed and poplar pollenkitts (see Section 3.2).

For pollenkitt and ammonium sulfate mixtures (panel (b) in Fig. 1), AS enhances CCN activity of ragweed, as might immediately be expected upon addition of a highly hygroscopic salt. However, for poplar, the enhancement of CCN activity is only seen for larger particles, whereas CCN activation is suppressed for smaller particles. In case of both mixtures, the slope of the $\ln D_{p,50}$ vs. $\ln SS$ plots change significantly compared to the respective pure pollenkitt particles. This indicates the presence of now significant size-dependent effects on CCN activity introduced by the interaction between pollenkitt and the inorganic salt via either solubility or surface tension effects.

## 3.2 Pollenkitt surface activity

**Table 3.** Pollenkitt surface activity: Szyszkowski fitting parameters according to Eq. (3) for bulk aqueous solutions of ragweed and poplar pollenkitts and their mixtures with ammonium sulfate.

| Sample | a [mN m$^{-1}$] | b [x10$^{-4}$g L$^{-1}$] | $\chi^2$ |
|---|---|---|---|
| Poplar | 3.67±0.244 | 1.49±0.543 | 3.640 |
| Ragweed | 3.48±0.207 | 1.93±0.609 | 2.316 |
| Poplar + 20 mass% AS | 3.53±0.206 | 0.180±0.0623 | 3.530 |
| Ragweed + 20 mass% AS | 3.37±0.156 | 0.230±0.0615 | 1.841 |

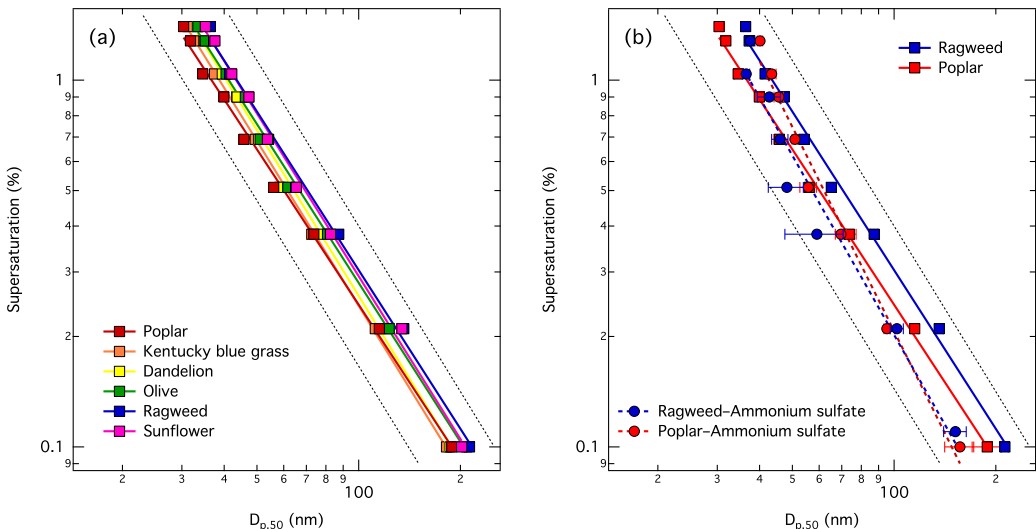

**Figure 1.** Measured CCN activity in terms of critical dry diameter ($D_{p,50}$) vs. supersaturation ($SS$) for particles comprising (a) six pure pollenkitts, and (b) mixtures of poplar and ragweed pollenkitt with 20% by mass of ammonium sulfate, shown together with results for the respective pure pollenkitt particles. Lines are fitted to the data to determine the slope in $\ln - \ln$ space. Dashed black lines have slope $-3/2$ in $\ln - \ln$ space and are shown on the graphs to guide the eye.

Measured surface tension for ragweed and poplar pollenkitts and their mixtures with ammonium sulfate are shown in Fig. 2. Error bars shown on the plot represent standard deviations calculated from independent measurements of three droplets. The resulting fitting parameters $a$ and $b$ in Eq. (3) are given in Table 3. Since these parameters have been obtained by fitting measurements for each solute composition separately, such that the relative ratio of pollenkitt and ammonium sulfate solute has not been varied independently, the resulting parametrizations are only pseudo-ternary. Implications of this are discussed below.

Both pollenkitts are moderately strong surfactants and able to reduce aqueous surface tension to values below 50 mN m$^{-1}$ at concentrations of about 0.1 g L$^{-1}$. Below, we infer pollenkitt average molecular masses using KTA to be 825–1130 and 400–550 g mol$^{-1}$ for ragweed and poplar, respectively. For molecular masses in the range 100–1000 g mol$^{-1}$, the pollenkitt concentrations where surface tension drops below 50 mN m$^{-1}$ correspond to 0.1–1 mM. The investigated concentration range covers those of stock solutions used for pollenkitt particle generation for CCN measurements. However, it is not possible to conclude that potential critical micelle concentrations for the solutions are also covered by the investigated range, i.e. that the measured surface tensions have plateaued for the highest concentrations studied here, or that such a plateau can be found.

We see that poplar pollenkitt is the stronger surfactant compared to ragweed, and in both cases that surface tension is decreased upon addition of ammonium sulfate compared to binary solutions with the same pollenkitt content. The latter ob-

servation indicates a modest salting out effect for the given PK:AS mixing ratio of 4:1 by mass, where ammonium sulfate increases pollenkitt aqueous activity and bulk-to-surface partitioning. Both observations suggest that surface activity might indeed affect pollenkitt CCN activity as well.

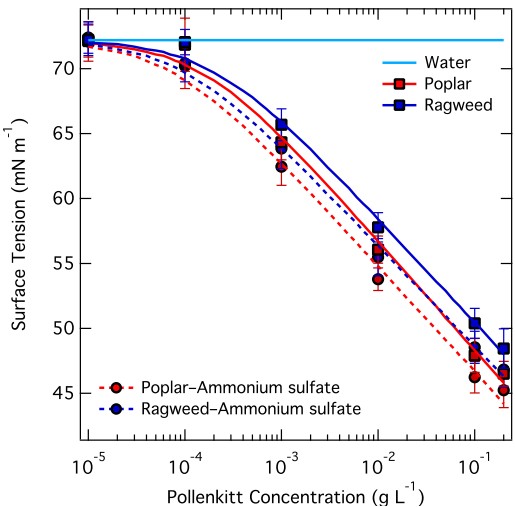

**Figure 2.** Measured surface tension of ragweed and poplar pollenkitts and their 20% by dry solute mass mixtures with ammonium sulfate, together with curves predicted by the parameterization of Eq. (3) and fitting parameter values given in Table 3.

### 3.3 Pollenkitt experimental $\kappa$ values

Measured CCN activity for the different pollenkitts and the mixtures of pollenkitt with ammonium sulfate were inverted according to the relations given in Section 2.6 for obtaining $\kappa$ values for each particle composition, size, and supersaturation state. Figure 3 shows size resolved experimental $\kappa$ values for pure dandelion, Kentucky bluegrass, olive, and sunflower pollenkitts in panels a)–d), and for pure poplar and ragweed pollenkitt, respectively, and their mixtures with ammonium sulfate, in panels e) and f). For each particle composition, average $\kappa$ values for the measured supersaturation and particle critical size range and the standard deviation for this average is also shown. In general, the particle hygroscopicity is moderate to low, with $\kappa$ values

for pure pollenkitts particles comparable to previous estimates of organic hygroscopicity from ambient measurements, which range from 0–0.25 (Shantz et al., 2008; Wang et al., 2008; Bougiatioti et al., 2009; Gunthe et al., 2009; Chang et al., 2010; Dusek et al., 2010; Cerully et al., 2011; Moore et al., 2012; Mei et al., 2013a, b; Cerully et al., 2015; Pöhlker et al., 2016; Thalman et al., 2017). We find pollenkitt to be more hygroscopic at supersaturated conditions than the whole pollen grains studied by Pope (2010) and Griffiths et al. (2012) at subsaturated humidities. This is consistent with pollenkitt contributing dis-

proportionately to overall pollen hygroscopicity, but may also be related to varying hygroscopicity between different humidity regimes, as was also noted by e.g. Kristensen et al. (2014) and Hansen et al. (2015). In our measurements, the average $\kappa$ values

for pure pollenkitt particles follow the same order as CCN activity at 1 % supersaturation, ragweed $\approx$ sunflower $<$ olive $<$ dandelion $<$ Kentucky bluegrass $<$ poplar (see Table 2).

For ragweed pollenkitt, mixing with ammonium sulfate significantly increases particle hygroscopicity, as expected from the higher intrinsic hygroscopicity of the inorganic salt, $\kappa_{AS} = 0.61$ (Petters and Kreidenweis, 2007). This is however not the case for poplar, where overall particle hygroscopicity is observed to be both higher and lower in ammonium sulfate mixtures, compared to pure poplar pollenkitt, and the average $\kappa$ value for mixtures is actually lower than for the pure pollenkitt. We return to this point in the following sections.

It is immediately evident from Fig. 3 that experimental $\kappa$ values for all pure pollenkitts and the two mixtures with ammonium sulfate depend on dry particle size. As the measured activation curves are consistent across supersaturations, we believe that this size dependency is real, rather than due to instrument artifacts. Furthermore, we note that the shape of the $\kappa$ size dependency is similar for all particle compositions. In particular, for all pure pollenkitts, the $\kappa$ values are stable at, or slightly below, the respective average values, for particle sizes above roughly 70 nm. For smaller sizes, $\kappa$ values first peak roughly between 50-70 nm, then steeply decrease, with possible further fluctuations, for still decreasing sizes. In several cases, the peak values are outside the rage of standard deviations for the average $\kappa$ values.

We suggest that this size variation in particle hygroscopicity reflects the impact of pollenkitt surface activity on CCN activation. Since $\kappa$ values are inverted under the assumption of constant pure water surface tension, any effect of surface tension reduction on CCN activity is included in the resulting $\kappa$ values. The relative diameter growth factor at the critical point of activation –the ratio of droplet size at activation to the dry particle size, here as $d_c/D_{p,50}$– decreases with particle size (Prisle et al., 2010b), such that smaller particles activate as more concentrated solutions with higher surface area-to-bulk volume ratios, compared to large particles. The size dependency in pollenkitt $\kappa$ values thus reflects the changing relative contributions of surface partitioning and surface tension reduction to the water uptake equilibrium. The smallest particles are depleted in bulk solute from bulk-to-surface partitioning, driven by the relatively large surface areas of small droplets. This effect is reducing $\kappa$ values to a greater extent than any simultaneous increase in effective $\kappa$ from reduced surface tension, even at the highest total solute concentrations in the droplets. For somewhat larger particles, the relative magnitudes of these two competing effects are reversed. The surface partitioning of pollenkitt is not longer strong enough to completely counter the increase in $\kappa$ values from surface tension reduction. As particles get even larger, activating droplets become sufficiently large and dilute that neither of these effects are very strong, and the size dependency of $\kappa$ values levels off.

For the pollenkitt–ammonium sulfate mixtures, the $\kappa$ size dependency looks qualitatively similar to that generally observed for the pure pollenkitt particles. However, the peak $\kappa$ value is shifted to smaller particles for ragweed and to larger particles for poplar, compared to the pure pollenkitts. As mentioned above, $\kappa$ values of ammonium sulfate mixtures with ragweed are higher than for the pure pollenkitt, while still lower than for pure ammonium sulfate, whereas $\kappa$ values for poplar–ammonium sulfate mixtures are significantly lower than for pure pollenkitt at small particle sizes, with a strongly decreasing trend for decreasing size. Again, we believe that these observations reflect the impact of pollenkitt surface activity on CCN activation.

As seen in Section 3.2, poplar pollenkitt is a stronger surfactant and, as we saw in Figs. 1 and 3, has a greater intrinsic CCN activity (smaller $D_{p,50}$, larger $\kappa$) than ragweed. For ragweed, we therefore see a stronger effect of salting out of pollenkitt by

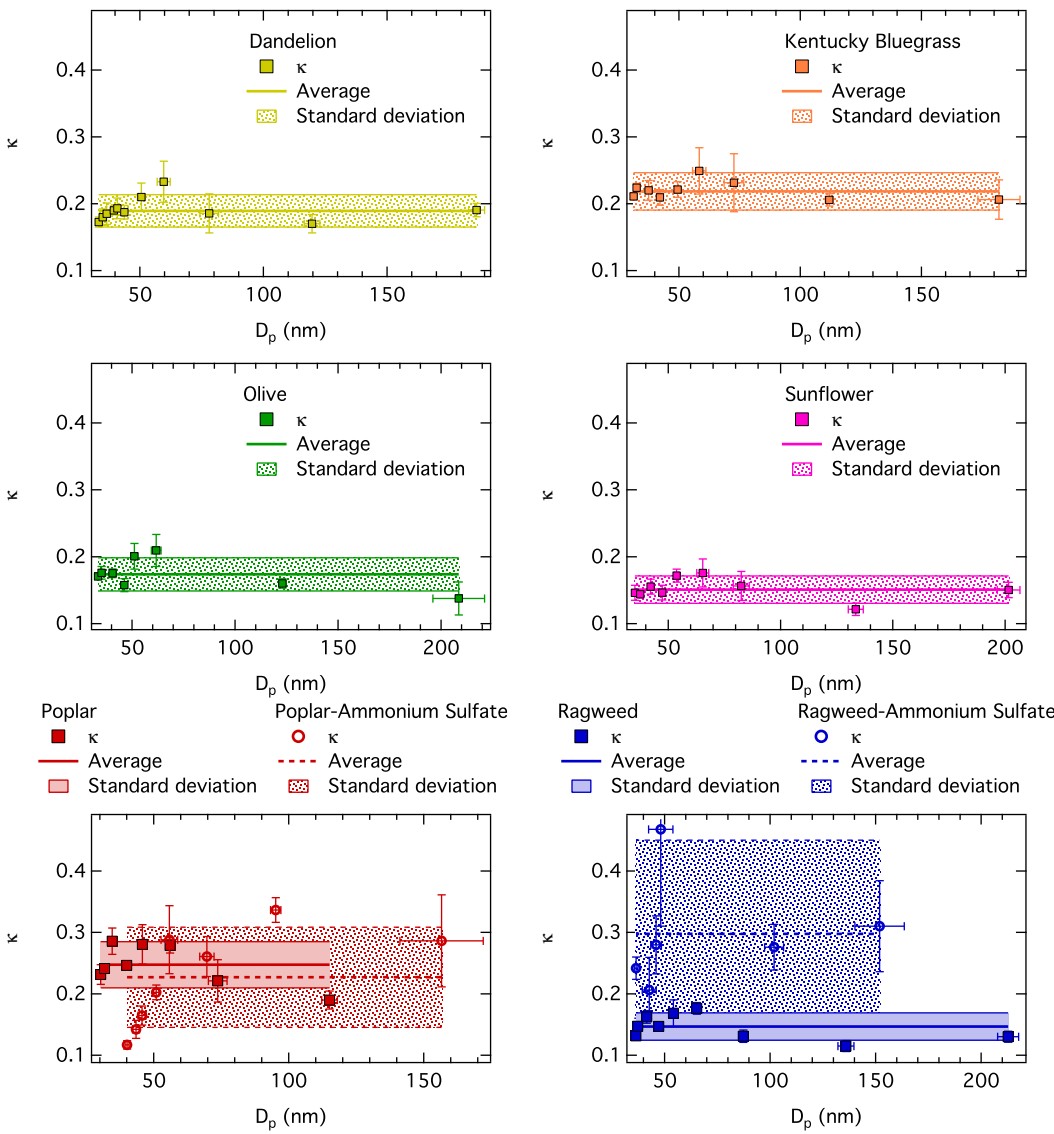

**Figure 3.** Hygroscopicity values $\kappa$ from CCN activity measured by SMCA for all particle compositions. Panels (a)–(d) for pure dandelion, Kentucky bluegrass, olive, and sunflower pollenkitt. Panels (e) and (f) for pure poplar and ragweed pollenkitt, respectively, and their mixtures with ammonium sulfate. Error bars are standard deviations from the $D_{p,50}$ calculated from the SMCA data inversion and standard deviations of the $\kappa$ values inverted from pairs of $D_{p,50}$ and $SS_c$.

ammonium sulfate at the most sensitive smallest and most concentrated activating droplets. First, $\kappa$ values increase steeply with size, and as concentrations become more dilute, both surface activity and partitioning depletion effects quickly taper off. For poplar pollenkitt, these effects and their interaction are evident over a wider range of particle sizes and thus corresponding activating droplet concentrations, since effects of both bulk depletion from surface partitioning and surface tension reduction of the more surface active pollenkitt remain significant for more dilute droplet compositions.

### 3.4  Effects of surface activity on predicted pollenkitt CCN activation

Experimental particle CCN activity and derived $\kappa$ values were compared to those obtained from different thermodynamic models, which account for effects of composition (assuming volume additivity) and surface activity, subject to different simplifying assumptions (Prisle et al., 2010b, 2011). The purpose is to test the applicability of these various models and their predictive strength for assessing overall hygroscopicity and cloud droplet activation potential for pollenkitts as examples of complex atmospheric OA mixtures. The Köhler theory framework is well-suited for gauging the nature of the surface activity effects on equilibrium water uptake to the aerosol phase. However, as mentioned previously, the underlying aqueous phase properties have implications for aerosol–water interactions beyond cloud microphysics.

#### 3.4.1  Thermodynamic Köhler models

Figure 4 shows predictions of CCN activity for pure and ammonium sulfate-mixed poplar and ragweed pollenkitt particles from the three thermodynamic Köhler models. In all cases, we have assumed a dry pollenkitt mass density of $1.2\ \mathrm{g\ cm^{-3}}$, which gave the best agreement with experimental CCN data over the density range $0.8-1.2\ \mathrm{g\ cm^{-3}}$.

Together with CCN activation, for each dry particle size and composition, we also evaluated the droplet surface tension specifically at the critical point of droplet activation, $d_c$. These results are shown in Fig. 5 for the same model runs as shown in Fig. 4. For both the pure pollenkitts, the full partitioning model typically predicts surface tensions in activating droplets which are about 10% reduced, compared to pure water. Model results are therefore shown here for the corresponding KTA-derived average effective molecular weights of $400\ \mathrm{g\ mol^{-1}}$ for poplar and $825\ \mathrm{g\ mol^{-1}}$ for ragweed pollenkitts, respectively, which are seen as the most internally consistent estimates. Over the range of KTA-derived pollenkitt molecular weights given in Section 3.2, corresponding to 0-10% reduction in droplet surface tension from pure water, all give essentially the same resulting CCN activity, reflecting the modest direct impact of variations in assumed average molecular weight on modeled CCN activity for the two pollenkitts. This is also reflected in the similarity of CCN activity predicted with the full partitioning model and the water model omitting any effects of pollenkitt surface activity.

For mixed PK–AS particles, droplet surface tension at activation is typically slightly higher than for the corresponding pure pollenkitts. This reflects both that activating droplets are more dilute, due to the higher hygroscopicity of AS mixtures, as well as a small contribution from ammonium sulfate itself on aqueous surface tension.

Returning to Fig. 4, we note that overall, neither of the three thermodynamic models is able to capture measured pollenkitt CCN activity well over the full particle size range studied. In particular, as observed in several previous studies for other atmospheric surfactants, the bulk model using droplet surface tensions based on total particle concentrations, and specifically

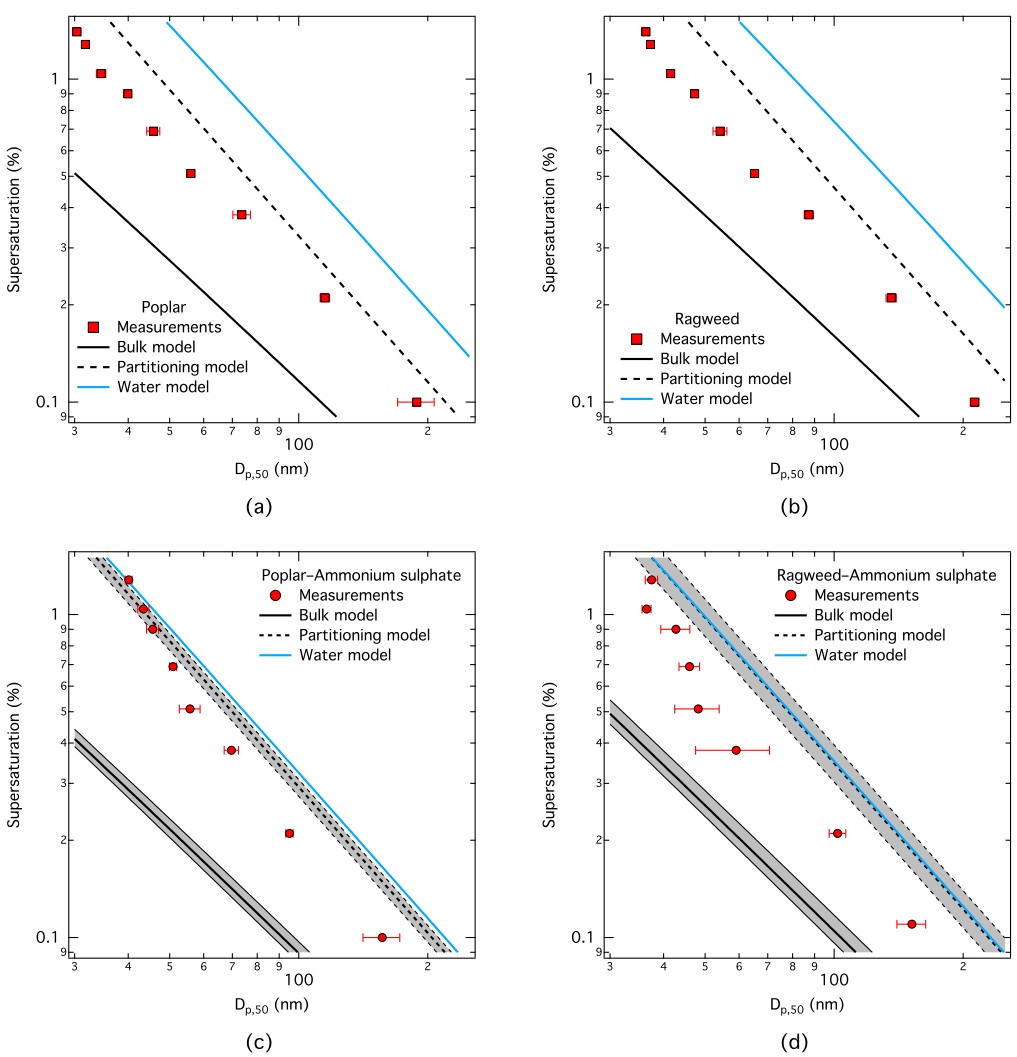

**Figure 4.** CCN activity modeled with the three different thermodynamic models (bulk, partitioning, and water) for particles comprising (a) pure poplar, (b) pure ragweed, (c) poplar and 20% by mass of ammonium sulfate, and (d) ragweed and 20% by mass of ammonium sulfate. For mixtures with ammonium sulfate, uncertainty ranges corresponding to a 10 wt% uncertainty in dry particle composition are indicated for the two models including concentration dependent surface tension, bulk and partitioning.

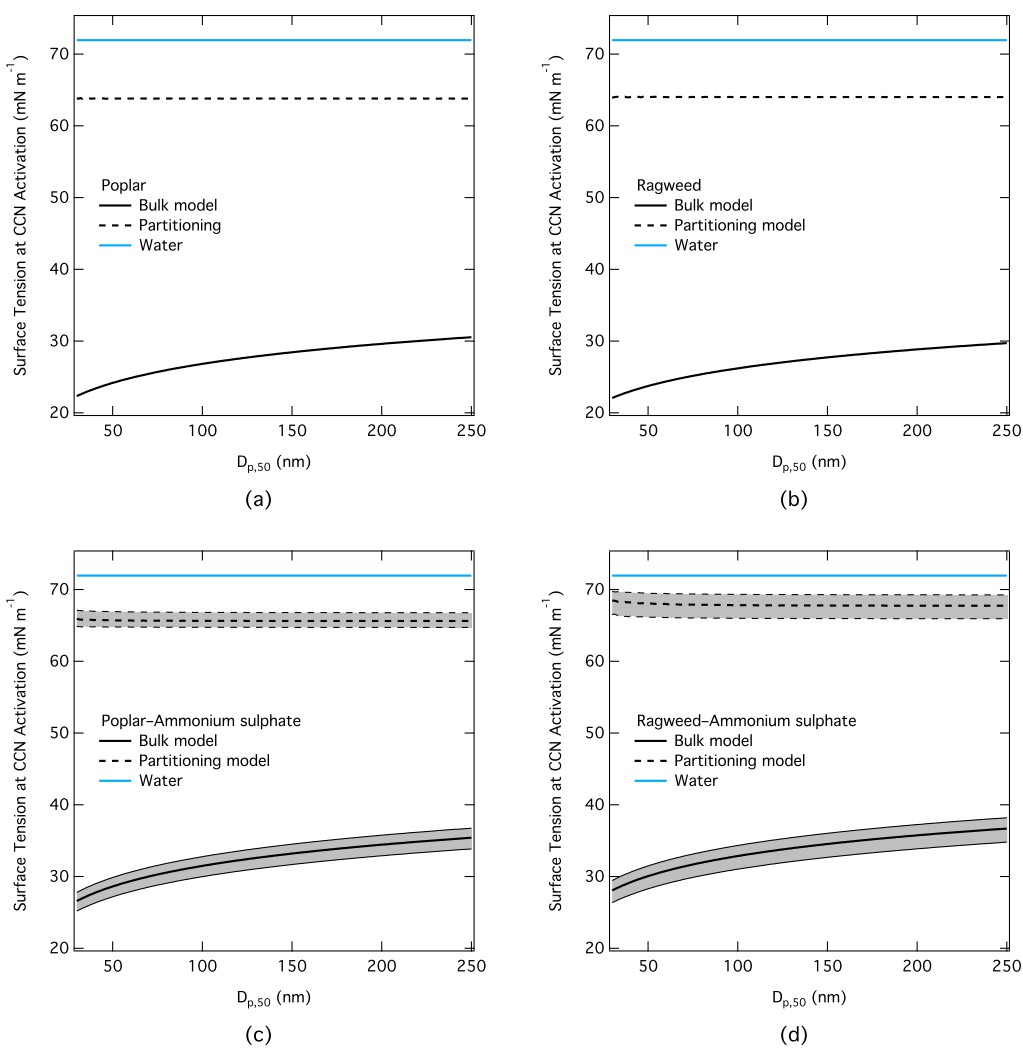

**Figure 5.** Droplet surface tension at the point of CCN activation modeled with the three different thermodynamic models (bulk, partitioning, and water) for particles comprising (a) pure poplar, (b) pure ragweed, (c) poplar and 20% by mass of ammonium sulfate, and (d) ragweed and 20% by mass of ammonium sulfate. For mixtures with ammonium sulfate, uncertainty ranges corresponding to a 10 wt% uncertainty in dry particle composition are indicated for the models including surface tension, bulk and partitioning.

omitting bulk composition changes from surface partitioning, greatly overestimates measured CCN activity in all cases (Prisle et al., 2008, 2010b; Kristensen et al., 2014; Hansen et al., 2015). However, our most comprehensive and thermodynamically consistent partitioning model is also not able to predict pollenkitt CCN activity across the full investigated particle size range. In previous work, the partitioning model was observed to represent surface active organic aerosol CCN activity well, especially for 2-3 component model aerosol systems comprising relatively strong and simple surfactants. Uncertainties in pollenkitt dry mass density and non-ideal aqueous activities, including unaccounted for synergy effects upon mixing with ammonium sulfate are likely contributing to this.

It has also been suggested that laboratory-generated surfactant particles from nebulization of an aqueous stock solution can be enriched in surfactant, compared to non-surface active co-solutes like ammonium sulfate. In Fig. 4, we show uncertainty ranges for mixed particle model results reflecting uncertainties in particle composition by a $\pm 10\%$ deviation in pollenkitt mass fraction. Even with such variations, we cannot reconcile any of the models with measured CCN activity. It is possible that the suggested enrichment could be even more dramatic, but its quantification and potential variation across dry particle sizes remains uncertain.

Both pollenkitts are seen in experiments to be more hygroscopic than predicted with the full partitioning model. The deviation is greater for pure pollenkitts and ragweed mixtures, compared to poplar mixtures, for which effects of bulk-to-surface partitioning are expected to be most pronounced, as poplar pollenkitt is the strongest surfactant and mixing with ammonium sulfate is seen to lead to salting out. In general, measured CCN activity falls between predictions from the two models which include aqueous surface tension reduction from the pollenkitt, suggesting that surface tension indeed impacts the CCN activity of pollenkitt. The overestimation of CCN activity by the bulk model also suggests that bulk depletion from surface partition plays a significant role, however, at the same time the underestimation of CCN activity by the full partitioning model suggests that the extent of bulk depletion may be overestimated in the current model description.

The water model, which treats pollenkitt like a regular solute and neglects specific effects of surface activity, significantly underestimates CCN activity of the pure pollenkitts but gives similar results to the full partitioning model for the mixed particles. This suggests that surface activity indeed plays a role in determining CCN activity of poplar and ragweed pollenkitt and that the overall effect of surface activity is to enhance pollenkitt CCN activity. In pure pollenkitt particles, the effect of surface tension is therefore seen to be stronger than that of bulk depletion. As noted in previous work (Prisle et al., 2011), the water model may predict CCN activity well for strong surfactants, where bulk-to-surface partitioning is more prominent. However, other properties of activating droplets, such as bulk and surface composition and critical droplet size, may not be similarly well represented by the simplistic description, as was recently verified experimentally by Ruehl et al. (2016). This may in turn have consequences for other aspects of atmospheric chemistry and aerosol-cloud-climate interactions (Prisle et al., 2012).

### 3.4.2 Simple partitioning

CCN activity predicted with the simple partitioning model are shown in Fig. 6. The simple model only gives meaningful results for mixed particles comprising a non-surface active hygroscopic components such as ammonium sulfate, because the hygro-

scopicity of surfactants is assumed to be vanishing. For the same reason, modeled results are identical for the two pollenkitts ragweed and poplar for a given assumed organic mass density regardless of the molecular mass of pollenkitt. In Fig. 6, we show results for three values of pollenkitt density over the range considered. We see that in all cases, the simple partitioning model strongly underestimates pollenkitt CCN activity, supporting the conclusion that droplet surface tension is indeed reduced at the point of CCN activation and also that pollenkitt has some intrinsic hygroscopicity, as further supported by the close similarity of predictions from both partitioning and water models. Perhaps counterintuitively, but as explained in detail in earlier work (Prisle et al., 2011), the simple representation works better for stronger surfactants. The pollenkitts are fairly surface active, but evidently not enough to be well represented by the simple model which was developed for even stronger surfactants, like SDS and fatty acid salts with straight aliphatic chains comprising 10 or more carbon atoms.

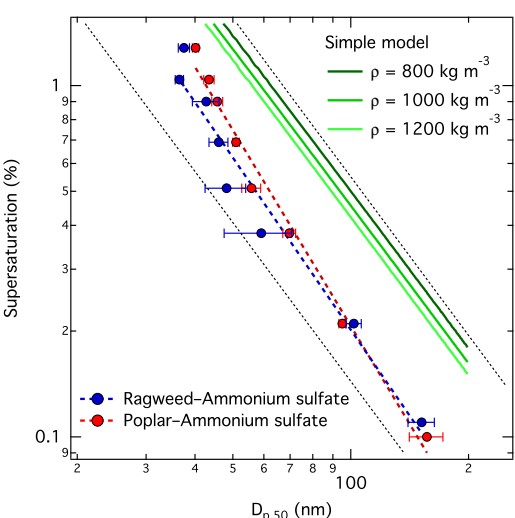

**Figure 6.** CCN activity modeled with the simple partitioning model for particles comprising pollenkitt and 20% by mass of ammonium sulfate. The simple model predicts the same CCN activation for all organics, given a specified dry organic mass density, as explained in the text. Model results are shown here for assumed pollenkitt densities of 0.8, 1.0, and 1.2 g cm$^{-3}$ together with measured CCN activation for poplar and ragweed mixtures with ammonium sulfate. Dashed black lines have slope $-3/2$ in $\ln - \ln$ space and are shown on the graphs to guide the eye.

### 3.4.3   Modeled $\kappa$ values

Experimental $\kappa$ values for poplar and ragweed pollenkitts and their mixtures with ammonium sulfate are compared to those obtained from predictions with the three different thermodynamic models in Fig. 7.

None of the models capture the shape of the size dependence of experimentally derived $\kappa$ values. The partitioning model $\kappa$ values hint of a very small decrease for the smallest particle sizes but not near the decrease observed in the experimental data. We believe this observation is caused by concentration dependent droplet non-ideality in combination with insufficient representation of the true ternary ammonium sulfate–pollenkitt aqueous phase interactions when using the pseudo-ternary surface tension parameterizations. These are based on a constant mixing ratio between ammonium sulfate and pollenkitt, whereas bulk-to-surface partitioning of the surfactant changes this ratio for the activating droplets. For highly pollenkitt-depleted smaller droplets, this ratio is significantly changed from the nominal 4:1 mixture, as discussed in detail by Prisle et al. (2010b, 2011). One consequence is that the partitioning depletion for small droplets is likely to be underestimated, compared to the larger droplets, in our present model results.

$\kappa$ values predicted with the bulk model increase steadily with decreasing dry particle diameter, reflecting how modeled surface tension of activating droplets decrease as these droplets get more concentrated. Bulk model $\kappa$ values are in all cases significantly greater than experimentally derived hygroscopicity. Interestingly, the predicted $\kappa$ values for mixed particles are greater than for pure ammonium sulfate for both pollenkitts, reflecting that effects of surface tension in the absence of partitioning depletion of the droplets would lead to higher effective hygroscopicity of the organic than even a highly hygroscopic inorganic salt. For poplar, this effect is predicted to be sufficiently pronounced to make even the intrinsic hygroscopicity of the pure pollenkitt particles larger than ammonium sulfate. For ragweed, the effect is seen for the smaller particles, which activate as more concentrated droplets.

By design, the water model predicts constant $\kappa$ values for particles under the ideal solution assumption applied here. Concentration-dependent water activity coefficients could potentially introduce a size dependency in $\kappa$ values as droplets dilute. However, sensitivity studies in previous work have shown that this effect is likely very small for the water model, where activating droplets are in general relatively more dilute than for the partitioning model (Prisle et al., 2010b). The close similarity of mixed pollenkitt–ammonium sulfate particle $\kappa$ values predicted with the partitioning and water models (for ragweed, these are essentially identical) underline the small overall effect of pollenkitt surface activity on droplet activation. The opposing influences of surface tension reduction and partitioning depletion yield resulting effective $\kappa$ values nearly identical to those predicted assuming no surface activity of pollenkitt. However, the difference between predictions for pure pollenkitt particles also indicate that this is partly due to the effect of the hygroscopicity of ammonium sulfate.

As we observed above, the strongest evidence for the presence of surface activity effects on CCN activation thermodynamics of pollenkitt comes from the deviation of the slope of $\ln D_{p,50}$ vs. $\ln SS$ from the ideal solution value of $-3/2$, especially seen for the more surface active poplar mixtures with ammonium sulfate. From the experimental $\kappa$ values, the size dependency in pollenkitt hygroscopicity is readily linked to surface activity through both surface tension and bulk-to-surface partitioning, as was also observed in previous work (Prisle et al., 2010b). Pollenkitt aqueous bulk solubility may play a role here as well. However, the primary mechanism would result in overestimation of CCN activity by the water model as is only seen for the smallest poplar–ammonium sulfate particles, which are also the most concentrated at the point of activation.

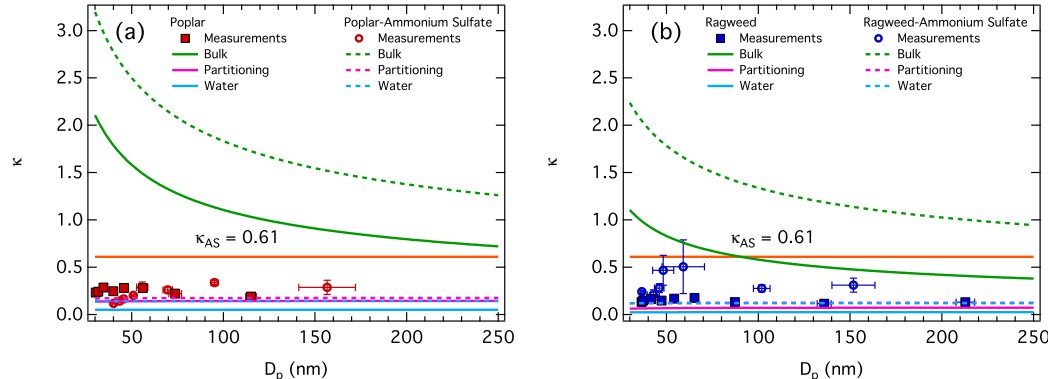

**Figure 7.** Hygroscopicity values $\kappa$ derived from modeled CCN activity for (a) poplar and (b) ragweed pollenkitt particles and their mixtures with ammonium sulfate. Also shown are the corresponding measurement derived hygroscopicity values and $\kappa$ value for pure ammonium sulfate for reference.

## 4 Modeling CCN activity of pollen grains and fragments

In the wider context of aerosol thermodynamics and cloud microphysics, pollenkitt is studied here as an example of atmospheric biogenic primary organic aerosol to demonstrate complex effects of organic surface activity on OA–water interactions. Specifically related to the atmospheric effects of pollen, results of this work can be applied to quantify the interaction of whole

pollen grains and their SSP fragments with ambient water. By resolving the specific water interactions in terms of the hygroscopicity parameter $\kappa$, we are able to decouple the contributions of bare (defatted) pollen grains or their fragments, soluble and surface active pollenkitt, and potential additional inorganic components such as ammonium sulfate formed by atmospheric processing of pollen. With the intrinsic hygroscopicity of each of these components constrained, the overall response of pollen grains and fragments can be modeled for a wide range of atmospheric conditions, including variations in humidity, tempera-

ture, pollen grain or fragment size and size distributions, and pollenkitt and secondary inorganic component mass fractions, without the specific knowledge of overall hygroscopicity from measurements performed at immediately matching conditions across the entire matrix.

To illustrate this, we calculate the variation of critical supersaturation with dry size of a single ragweed pollen grain or fragment. For simplicity, we here assume that both grains and fragments are spherical and composed of an insoluble core with

density of $\rho_{\text{insol.}} = 1.28$ g cm$^{-3}$, corresponding to the value measured at 100% RH (Harrington Jr. and Metzger, 1963), and a pollenkitt coating with the mass fraction of $W_{p,\text{PK}} = 0.15$ as measured by Lin et al. (2013) and density of $\rho_{\text{PK}} = 1.2$ g cm$^{-3}$

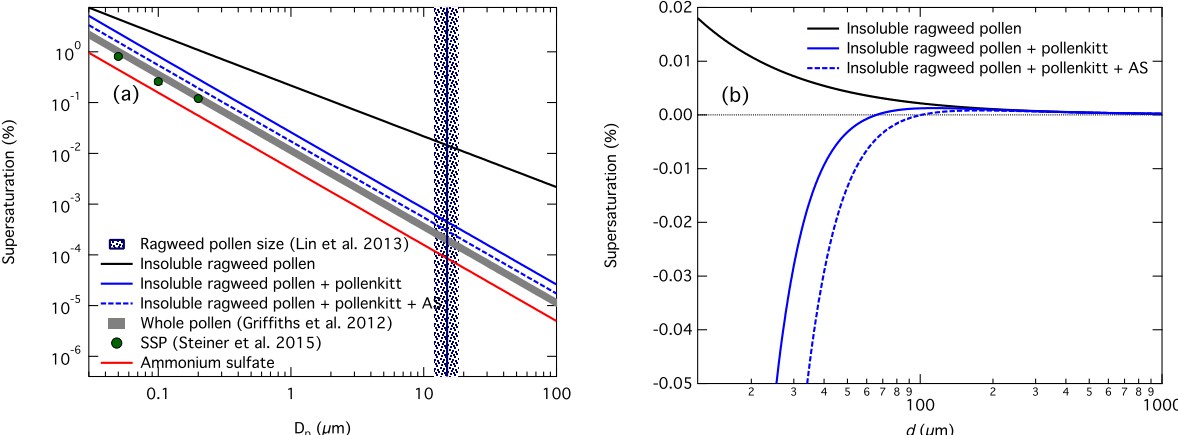

**Figure 8.** (a) Modeled CCN activity in terms of dry particle diameter ($D_p$) vs. critical supersaturation ($SS_c$) for ragweed pollen whole grains or fragments with and without contributions from pollenkitt and ammonium sulfate. CCN activity derived from hygroscopicity values for pure ammonium sulfate and a range of values measured for different whole pollen grains by Griffiths et al. (2012), and for SSP measured by Steiner et al. (2015), are shown for comparison; (b) Modeled Köhler growth curves for an initially dry 15 $\mu$m ragweed pollen grain (indicated as vertical line in the left panel) with the corresponding contributions from pollenkitt (15% mass fraction) and ammonium sulfate (in 1:4 mass ratio relative to pollenkitt).

as used in this work. Assuming volume additivity of the insoluble core and pollenkitt coating according to Eq. 2, the volume fraction of pollenkitt $\epsilon_{\mathrm{PK}}$ in the pollen grain is related to its mass fraction by

$$\epsilon_{\mathrm{PK}} = W_{p,\mathrm{PK}} \left( W_{p,\mathrm{PK}} + W_{p,\mathrm{insol.}} \frac{\rho_{\mathrm{PK}}}{\rho_{\mathrm{insol.}}} \right)^{-1} \tag{5}$$

where the insoluble core and pollenkitt mass and volume fractions are related as $W_{p,\mathrm{insol.}} = 1 - W_{p,\mathrm{PK}}$ and $\epsilon_{\mathrm{insol.}} = 1 - \epsilon_{\mathrm{PK}}$.

5 From this, the overall effective hygroscopicity $\kappa_{\mathrm{pollen}}$ of each ragweed pollen grain or fragment can be calculated from

$$\kappa = \sum_i \epsilon_i \kappa_i \tag{6}$$

and assuming individual hygroscopicity values of $\kappa_{\mathrm{insol.}} = 0$ for the insoluble core and $\kappa_{\mathrm{PK}} = 0.14$ for ragweed pollenkitt as derived from our measurements here. Critical supersaturations $SS_c$ are then calculated from $\kappa_{\mathrm{pollen}}$ as a function of pollen grain or fragment diameter $D_p$ according to Eq. 10 from Petters and Kreidenweis (2007):

10 $$SS_c = \left( \exp \left[ \sqrt{\frac{4A^3}{27 D_p^3 \kappa}} \right] - 1 \right) \times 100\% \tag{7}$$

where

$$A = \frac{4\sigma M_w}{RT\rho_w} \tag{8}$$

With a fixed mass fraction of pollenkitt, the volume fractions of each component are also constant according to Eq. 5. The overall ragweed pollen grain hygroscopicity $\kappa_{\text{pollen}}$ is therefore not a function of pollen grain diameter under these assumptions.
If $W_{p,\text{PK}}$ would vary across a pollen grain or fragment size distribution, volume fractions and $\kappa_{\text{pollen}}$ would vary accordingly.

    Following a similar procedure, we can add ammonium sulfate to the ragweed pollen grains and fragments, to simulate effects of atmospheric processing. If we assume a fixed relative mass ratio of $0.15 : 0.85$ between pollenkitt and the insoluble core, and that the mass ratio between ammonium sulfate and pollenkitt is $1 : 4$, as in this work, this yields the relations $W_{p,\text{insol.}} = \frac{0.85}{0.15}W_{p,\text{PK}}$ and $W_{p,\text{PK}} = 4W_{p,\text{AS}}$ between the mass fractions of individual pollen components, from which
each mass fraction can be determined under the constraint that $W_{p,\text{insol.}} + W_{p,\text{PK}} + W_{p,\text{AS}} = 1$. As for the binary case, with the assumption of volume additivity, each set of mass fractions yields invariant volume fractions of insoluble core, pollenkitt, and ammonium sulfate components. The combined volume fraction of soluble pollenkitt–ammonium sulfate mixture, $\epsilon_{\text{PK-AS}} = \epsilon_{\text{PK}} + \epsilon_{\text{AS}}$, is then determined from Eq. 5 by substituting $\epsilon_{\text{PK-AS}}$ for $\epsilon_{\text{PK}}$, the combined pollenkitt–ammonium sulfate mass fraction $W_{p,\text{PK-AS}} = W_{p,\text{PK}} + W_{p,\text{AS}}$ for $W_{p,\text{PK}}$, and the pollenkitt–ammonium sulfate mixture density $\rho_{\text{PK-AS}}$, as given
from Eq. 2 with $\rho_{\text{AS}} = 1.769 \text{ g cm}^{-3}$, for $\rho_{\text{PK}}$. With the volume fractions $\epsilon_{\text{PK-AS}}$ and $\epsilon_{\text{insol.}} = 1 - \epsilon_{\text{PK-AS}}$, the overall pollen grain hygroscopicity value $\kappa_{\text{pollen}}$ is then again evaluated from Eq. 6, now using $\kappa_{\text{insol.}} = 0$ for the insoluble core and the average $\kappa_{\text{PK-AS}} = 0.28$ determined in this work for the ragweed pollenkitt–ammonium sulfate mixture. Had the hygroscopicity of the PK-AS mixture not been explicitly known, the volume fractions of insoluble core, pollenkitt, and ammonium sulfate could have been determined individually, and $\kappa_{\text{pollen}}$ obtained from Eq. 6 with values of $\kappa_{\text{insol.}}$ and $\kappa_{\text{PK}}$ as before, and $\kappa_{\text{AS}} = 0.61$ for
pure ammonium sulfate. The critical supersaturation $SS_c$ is then again calculated with $\kappa_{\text{pollen}}$ from Eq. 7.

    Results of these simple calculations are shown in Figure 8. The left panel shows modeled critical supersaturations for pollen grains and fragments of varying sizes between 50 nm and 100 $\mu$m in diameter, assuming bare insoluble pollen grains, ragweed pollen grains with 15% by mass of pollenkitt, and finally ragweed pollen grains with the same pollenkitt–insoluble core mass ratio and addition of ammonium sulfate such that the PK–AS mass ratio is 4:1. For comparison is also shown the CCN
activity predicted across the same particle size range from Eq. 7 for pure ammonium sulfate with $\kappa_{\text{AS}} = 0.61$, for the range of subsaturated hygroscopicities between $0.08 - 0.17$ measured for whole pollen grains of different species by Griffiths et al. (2012), and critical supersaturations for SSP particles of 50, 100, and 200 nm measured by Steiner et al. (2015). The average whole ragweed pollen grain diameter of 15 $\mu$m reported by Lin et al. (2013) is also shown for reference (vertical line with uncertainty range). In the right panel, we show the calculated Köhler curves representing each of the assumed compositions,
bare insoluble core, insoluble core with pollenkitt, and with pollenkitt and AS, for the case of a 15 $\mu$m ragweed pollen grain. The growth curves illustrate the effect of pollenkitt hygroscopicity and the influence of AS on the overall cloud droplet activation process. Similar curves are obtained for each of the grain or fragment sizes shown in the left panel.

We see that even with the very simple assumptions applied here, the overall hygroscopicity derived from our measured CCN activity of ragweed pollenkitt agrees well with previous measurements for both whole pollen grains and SSP, especially considering the wide variation in size, species, and likely morphology. Our additive framework yields somewhat higher critical supersaturations (lower CCN activity) for binary pollen compositions (solid blue line) than the measurements of both subsatu-

rated whole pollen grain hygroscopicity by Griffiths et al. (2012) and supersaturated SSP by Steiner et al. (2015). The addition of ammonium sulfate to ragweed pollenkitt increases the overall predicted CCN activity (dashed blue line), but still not quite enough to reconcile the measurements for either whole pollen grains or SSP. This indicates that the assumption of vanishing hygroscopicity for the defatted, bare pollen grain is likely not entirely valid. We speculate that the absorption of water into the insoluble structure, eventually leading to pollen rupture as described by Steiner et al. (2015), may also contribute to a change

in pollen size which is captured in the subsaturated hygroscopic response measured by Griffiths et al. (2012). Defatted, bare pollen grains exhibit a wide range of ornamental features, as shown by Lin et al. (2013, 2015) for the pollens studied here. The simple spherical particle morphology assumed here therefore most likely does not fully represent the Kelvin effect on water uptake properties for these pollen grains and more detailed descriptions may need to take the specific shapes of pollen grains or their fragments into account.

The comparison to predictions from measurements by Griffiths et al. (2012) may also be affected by differences in concentration-dependent non-ideality, surface tension, and bulk-to-surface partitioning effects for soluble material between the sub- and supersaturated humidity regimes, which may affect pollenkitt hygroscopicity analogously to the concentration effects introduced by particle size and growth factor at activation seen in our present measurements. Here, we have for simplicity assumed a constant hygroscopicity parameter with pollen grain or fragment size, somewhat contrary to the findings of this work. For

larger grains, both the overall Kelvin effect and therefore the potential impact of surface tension, as well as the surface-to-bulk ratio, become smaller and the size dependency of particle hygroscopicity is therefore likely less pronounced than in our CCN measurements of particles comprising only pollenkitt and possibly ammonium sulfate.

## 5 Conclusions

We studied the surface tension and cloud forming potential of pollenkitts obtained from pollens of six different plant species,

as an example of biogenic primary organic aerosol, a class of atmospheric complex OA which has not previously been subject to detailed analysis of the role of surface activity in cloud microphysics. Our measurements demonstrate that pollenkitt is both moderately hygroscopic and surface active. Considering in particular the size dependency of CCN activity and the related hygroscopicity parameter $\kappa$, we see that surface activity indeed does impact pollenkitt cloud droplet activation thermodynamics. Our most detailed model calculations show that surface tension of activating droplets is moderately reduced but not to the

dramatic degree as has been suggested in studies of other biologically related surfactants (e.g. Noziere et al., 2014; Ekström et al., 2010). The overall tendency is for surface activity to enhance pollenkitt hygroscopicity, in line with recent findings of Ruehl et al. (2016) for the hygroscopicity of secondary organic aerosol and Ovadnevaite et al. (2017) for primary organic aerosol in a coastal marine environment. Contrary to previous work by Prisle and co-workers, which focused on stronger

or chemically less complex surfactants, we here see evidence of both surface tension reduction in activating droplets, and a size dependency reflecting complex interactions between a partitioning depletion (Raoult) effect and the surface tension (Kelvin) effect. We also find evidence for complex aqueous phase solute-solute interactions significantly impacting the size dependent hygroscopicity of pollenkitts and their mixtures with ammonium sulfate salt via bulk-to-surface partitioning and its enhancement from salting out. For cloud microphysics, these results add to an emerging picture of a role of surface active organic aerosol in Köhler theory which is complex and remains to be fully resolved.

Understanding both the intrinsic hygroscopicity of pollenkitts from different species and effects of their interactions with soluble aerosol components is a critical step for resolving and describing the specific impact of surface active pollenkitt on water interactions of pollen grains and their fragments in the atmosphere, including, but not limited to their impact on clouds. Purely on its own, we find that pollenkitt tends to exhibit hygroscopic uptake properties that are similar to other complex organic systems, such as SOA that have a effective hygroscopicity somewhere between 0.1 and 0.2 (Andreae and Rosenfeld, 2008; Jimenez et al., 2009). This means that per unit mass of pollenkitt, the water uptake does not vary considerably between pollenkitts, so that the pollen uptake of water is driven primarily by the mass fraction of pollenkitt on the pollen particles, which is species-specific (Pacini and Hesse, 2005), possibly in combination with pollen grain or fragment size and curvature of any local features. Understanding the specific role of pollenkitt is a key step in enabling predictions of pollen interactions with ambient water across a variety of conditions in the atmosphere, including distributions of pollen grain or fragment sizes, shapes, and amount of pollenkitt, possible interactions with secondary (inorganic and other) aerosol, and growth and activation responses to a range of different humidity regimes. Including such processes in atmospheric frameworks with specific account of pollenkitt aqueous thermodynamics may therefore contribute to establish the overall significance of pollen and SSP as a source of biogenic CCN and IN on regional and global scales and is currently the focus on continued model development.

A further implication of our present results for atmospheric processing of pollen is that, depending on the species and humidity regime, the pollenkitt may have a considerably different response to condensation of sulfate, nitrate, and other highly hygroscopic species. The response of pollen to atmospheric aging as a result of the different water uptake properties may therefore be even more diverse than expected from fresh pollenkitt. Apart from affecting the cloud droplet forming potential, secondary aerosol formation, and surface chemistry of pollen particles suspended in the atmosphere, the different response in water uptake also has potentially profound implications for the adhesive and rupture characteristics of pollen, and thus for biological and transport processes (Lin et al., 2015). Furthermore, understanding the pollenkitt-water interactions for pure and mixed pollenkitts may provide important insights to responses in allergenicity and biological function of pollenkitt in polluted and pristine environments.

*Data availability.* Experimental and modeled data underlying figures can be accessed by request from N. L. Prisle (nonne.prisle@oulu.fi) and A. Nenes (athanasios.nenes@gatech.edu).

*Author contributions.* N.L.P., A.N., and J.C.M. designed experiments, N.L.P. designed and performed all model simulations. H.L. made the samples and performed surface tension measurements, and S.P. and J.J.L. performed CCN measurements. N.L.P. and J.J.L. analyzed the data and wrote the paper.

*Competing interests.* The authors declare no competing interests.

5  *Acknowledgements.* N.L.P. gratefully acknowledges the funding received from the European Research Council (ERC) under the European Union's Horizon 2020 research and innovation programme, Project SURFACE (Grant Agreement No. 717022). N.L.P. and J.J.L. are also grateful for the financial contribution from the Academy of Finland (Grant Nos. 308238, 314175, 290145, and 257411). A.N. acknowledges support from the European Research Council Project PyroTRACH (Grant Agreement no. 726165), a Georgia Power Faculty chair and a Cullen-Peck Fellowship from the Georgia Institute of Technology.

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
