# Peer review of "CCN activity of six pollenkitts and the influence of their surface activity."

_Atmospheric Chemistry and Physics, 2018_

## Referee Comment (RC1) · Anonymous Referee #1 · 23 Jun 2018

Overview:

The authors well executed a well written study to assess the cloud-forming propensity and water uptake capacity of various species of pollenkitt, a viscous substance that encapsulates pollen particles, in both binary solution and ternary mixtures including ammonium sulfate (AS). By measuring the particle surface tension and critical point, and deriving hygroscopic parameter kappa (k), for these mixtures they conclude that the surface active nature of pollenkitt influences the condensation nuclei (CN) activity and the size-dependent k via both bulk-surface partitioning and surface tension depression. Their results indicate that the surface activity of pollenkitt result in heightened hygroscopicity and in some cases a salting out effect caused by the presence of AS in the ternary mixtures increases the bulk to surface partitioning. As briefly touched

upon by the authors in the introduction, the prevalence of pollen(kitt) in proximity to real world cloud-forming meteorologies is uncertain. Even granting that submicron subpollen particles SSP are found at altitudes relevant for cloud formation, there is no reason to believe that the systems studied, pure pollentkitt and pollenkitt + AS (i.e. no bulk pollen grain material), would be present. Much of the studys motivation and contextualisation is predicated on cloud forming potential of the pollenkitt so it is unfortunate that the link to real world cloud formation remains tenuous, thereby diminishing the impact of the study.

Specific comments:

$D_{p,50}$ often referred to as a critical diameter throughout the study should be referred to as the dry activation diameter or similar. It is easy to confuse this with the critical point of the köhler curve, and the change will make terminology consistent with Moore et al. 2010.

It would be useful to see Figure 2 plotted against particle size, along with the köhler curves as predicted by the four models.

Page 2, line 28: Is the data available to quantify these high humidities? Or at least specify sub- or supersaturated?

Page 3, line 32: Ovadnevaite et al study better characterised as a coastal rather than oceanic location. Furthermore, the aerosol size distribution and chemical composition they studied is not representative of marine aerosol at large – I suggest removing "with potential global implications due to the relative significance of marine aerosol". See Heintzenberg et al (2000), for example, for comparison.

Page 3, line 6: Around here I suggest adding the reference Lowe et al. (2016) – uses 4 surfactant models, very similar conceptually to those in this study, to model CCN spectra across a similar range of supersaturations.

Page 4, line 32: State that this is $D_{p,50}$

[Figure]

Page 4, line 34: State somewhere the numerical values of all supersaturation bands.

Page 5, line 7: Is there some rationale for choosing this mass fraction? I guess to be a representative aerosol composition in a vegetation-rich region. I think it would be beneficial to add some motivating text. Furthermore, it would be have been nice to see how the results and conclusions might change subject to a varying mass fraction. I recommend including something to this effect in the revisions if possible.

Page 7, line 19: Is c_PK the total concentration for the bulk model and surface phase concentration in the partitioning model? If so, would be good to clarify here.

Page 8, section 2.5: state what you are assuming for surface tension for this procedure. If sigma=sigma_w then change in Eq. 1. Otherwise the reader doesn't get this information until page 12 line 19.

Page 8, line 12: Is this Dp,c the same as Dp,50? C.f. previous comments regarding activation and critical sizes.

Page 19, line 15: Do you mean Ragweed rather than poplar? The solid green curve for poplar is > k_AS for all sizes.

Technical comments:

Cloud activation is strange terminology. I suggest changing to condensation nuclei (CN) activation, or similar, throughout

Page 5, line 26: remove "(increasing d)"

Page 8, Eq. 4: Replace M_w and rho_w with partial molar volume as in Eq.1. Then no need to redefine. R also already defined under Eq.1

Page 9, line 11: section 3.2 ?

Page 11, line 5: change weight –> mass

Page 12, line 20: Define growth factor

Page 12, line 25: "high droplet total concentration", a bit confusing replace with "high total pollenkitt concentrations" or similar (if that's what you mean).

Page 12, line 28: nether -> neither

Figure 3: explain box whiskers in caption

References:

Heintzenberg, J., Covert, D., and Van Dingenen, R.: Size distribution and chemical composition of marine aerosols: a compilation and review, Tellus B, 52, 1104–1122, doi:10.1034/j.1600- 0889.2000.00136.x, 2000

Lowe, S., Partridge, D. G., Topping, D., and Stier, P.: Inverse modelling of Köhler theory – Part 1: A response surface analysis of CCN spectra with respect to surface-active organic species, Atmos. Chem. Phys., 16, 10941-10963, https://doi.org/10.5194/acp-16-10941-2016, 2016.

Moore, R. H., Nenes, A., and Medina, J.: Scanning Mobility CCN Analysis–A Method for Fast Measurements of Size-Resolved CCN Distributions and Activation Kinetics, Aerosol Science and Technology, 44, 861–871, https://doi.org/10.1080/02786826.2010.498715, 2010

Ovadnevaite, J., Zuend, A., Laaksonen, A., Sanchez, K. J., Roberts, G., Ceburnis, D., Decesari, S., Rinaldi, M., Hodas, N., Facchini, M. C., 15 Seinfeld, J. H., and O' Dowd, C.: Surface tension prevails over solute effect in organic-influenced cloud droplet activation, Nature, 546, 637–641, 2017.

---

## Referee Comment (RC2) · Anonymous Referee #2 · 6 Jul 2018

The paper deals with laboratory measurements of hygroscopicity and CCN activity of particles obtained by the aerosolization of a biological material – the pollenkitt – which is found in nature as coating of pollen grains. The results provide convincing evidence of surface tension effects on the CCN activation of submicrometric particles obtained by pollenkitt. The experimental data, interpreted on the basis of Koehler theory analysis, indicate that the partitioning of surface-active organic substances of pollenkitt reduce but not cancel the surface tension depression in activating cloud droplets. I found no errors in the methodology. My main concern is instead about the actual impact of these results: is that simply that pollenkitt aerosols are proved to be good CCN? Or that the interactions between organic compounds of pollenkitt with water have implications for the allergenicity of pollen? Or again is the emphasis on the fact that it was possible

to observe surface tension effects in CCN nucleation studies using the Koehler theory analysis, contrary to the first studies introducing the partitioning model? Depending on the actual focus of the paper, the Authors should provide a more systematic comparison with the literature. If the fragmentation of pollen grains is actually a source of pollenkitt aerosols in the atmosphere, as suggested by the work of Steiner et al. (2015), in what kind of environments this process can actually provide a significant contribution to CCN concentrations? Is this study relevant for representing CCN formation in the pre-industrial atmosphere? Or in certain pristine regions in the tropics, like the Amazon basin, where new particle formation does not occur and the generation of new submicron particles can be regulated by primary biological emissions (Poehlker et al. Science, 337, 10.1126/science.1223264, 2012)? In summary, the impacts of the present study should be more clearly stated by providing an appropriate context.
* * *

---

## Author Response (AR1)

**Author response to reviewers' comments**

Nønne L. Prisle[1,2,3,4], Jack J. Lin[1,3], Sara Purdue[3], Haisheng Lin[4], J. Carson Meredith[4], and Athanasios Nenes[3,4,5,6]

[1]University of Oulu, Nano and Molecular Systems Research Unit, P.O. Box 3000, 90014, University of Oulu, Oulu, Finland
[2]University of Helsinki, Department of Physics, P.O. Box 64, 00014, University of Helsinki, Helsinki, Finland
[3]Georgia Institute of Technology, School of Earth & Atmospheric Sciences, 311 Ferst Drive, Atlanta, GA 30332, USA
[4]Georgia Institute of Technology, School of Chemical & Biomolecular Engineering, 311 Ferst Drive, Atlanta, GA 30332, USA
[5]Institute of Chemical Engineering Sciences (ICE-HT), Foundation for Research, Patras, 26504, Greece
[6]Institute for Environmental Research and Sustainable Development, National Observatory of Athens, 15236, Athens, Greece

**Correspondence:** N. L. Prisle (nonne.prisle@oulu.fi)

We thank both reviewers for their careful revision of our manuscript, keen eyes on detail, and thoughtful comments.

A common point made by both reviewers relate to the atmospheric implications and impact of the work. Our main motivation for the work has been from the point of cloud microphysics, to evaluate the different Köhler frameworks including surface activity effects for a new type of atmospheric organic aerosol (OA) mixture. There is likely a great variety of surface active OA in the atmosphere, but the diversity of systems which have been analyzed for detailed surfactant effects on cloud microphysical processes is still rather narrow in terms of molecular complexity and compound classes studied (Petters and Petters, 2016). As we here observe a somewhat different hygroscopicity response to OA surface activity from previous work, supporting recent atmospherically relevant observations, more such investigations seem both warranted and timely. Pollenkitt is an example of atmospheric complex biogenic primary OA which is a class of OA that to our knowledge has not previously been subject to such analysis. Pollenkitt was especially well-suited, as extraction of the pollenkitt mixture from collected pollen grains allowed us to obtain sufficient amounts of sample to perform both the cloud condensation nuclei (CCN) and surface tension analysis in parallel, a common challenge for the type of studies in question and a key factor governing the scarcity of studies made for atmospheric OA so far. Furthermore, pollenkitt was interesting due to its unresolved chemical composition and higher aqueous solubility compared to most previously studied systems. It is possible that pollenkitt has similarities to other water soluble surface active biogenic OA, but that remains to be further established. Mixing with ammonium sulfate salt in the aqueous phase has atmospheric relevance considering aging processes in various environments, including polluted air masses and cloud processing, but is also used here as a physico-chemical indicator of surface activity effects, highlighting the magnitude of characteristic signatures of OA aqueous surface activity.

The context and intended impact of this work therefore lies primarily in challenging and widening our understanding of atmospheric surface active OA and specifically how surface activity impacts cloud microphysics, which remains one of the key parameters introducing uncertainties in predicted CCN and cloud droplet number concentrations in the atmosphere (Prisle et al., 2012; Lowe et al., 2016; Forestieri et al., 2018, and references therein). Specifically for pollen, our present results contribute to laying out the foundation for a more general and comprehensive treatment of atmospheric effects by enabling modeling

of hygroscopic water uptake and cloud formation across a wide range of conditions, e.g. humidity regimes and particle size distributions, accounting for individual sizes and shapes of pollen grains and fragments, fractions of pollenkitt and possible presence of inorganic secondary aerosol. Pollen comes in a great variety of shapes and sizes (Lin et al., 2013, e.g.). A decoupled understanding of contributions from different pollen components is therefore essential for modeling water interactions across a wide range of atmospheric conditions, which may not all be feasible to reproduce in controlled experiments. Such a treatment will require further model development and is the focus of ongoing efforts in collaboration with the regional and global modeling community. In addition to providing information and enabling modeling of the cloud forming potential of pollen grains and fragments, CCN activity analysis is also simply a means to characterize water interactions of small amounts of sample material. Such interactions are critical to the role of pollenkitt in transportation and biological functions of pollen, but a detailed study of the impact of our results on these processes lies outside the scope of this work.

We have modified the abstract, introduction and conclusions sections of our manuscript to clarify these points, and added an example (Section 4) illustrating how the specific pollenkitt hygroscopicity determined in this work can be used for predictions of overall CCN activity for pollen grains of different sizes, with and without the presence of ammonium sulfate.

Below we respond to specific comments from **Referee #1** in a point-wise fashion.

Specific comments:

1. Dp,50 often referred to as a critical diameter throughout the study should be referred to as the dry activation diameter or similar. It is easy to confuse this with the critical point of the köhler curve, and the change will make terminology consistent with Moore et al. 2010.

   We now refer to $D_{p,50}$ as critical dry diameter throughout the manuscript, following the terminology of e.g. Rose et al. (2010) and Kristensen et al. (2014).

2. It would be useful to see Figure 2 plotted against particle size, along with the köhler curves as predicted by the four models.

   Figure 2 shows measurements of bulk solution surface tension made from pendant drops that were not varied in size, so it is not possible to plot Fig. 2 against particle size along with the Köhler curves. The pendant drops are large enough to be considered purely macroscopic (bulk) solutions and therefore (unfortunately!) do not contain any information of size dependent surface tension. In case the reviewer refers to the evolution of modeled concentration dependent surface tension with dilution of a droplet and the impact on the shape of the Köhler curve, this has been shown in several previous studies, e.g. Shulman et al. (1996), Sorjamaa et al. (2004), Prisle et al. (2008), Prisle et al. (2010), and Prisle et al. (2011), and for the sake of brevity we therefore focus here on the variation of predicted droplet surface tension at activation across different dry particle sizes and compositions.

3. Page 2, line 28: Is the data available to quantify these high humidities? Or at least specify sub- or supersaturated?

   Yes, the humidities are available. The text has been updated with references to the experiments and now reads:

"Steiner et al. (2015) characterized the cloud droplet forming potential (CCN activity) of so-called submicron subpollen particles (SSP), which form as fragments from whole pollen grains. Laboratory experiments found whole pollen grains can rupture and release SSP when wetted by direct contact with liquid water or exposure to high ambient relative humidities of 80–96% (Grote et al., 2001; Taylor et al., 2002; Grote et al., 2003; Taylor et al., 2004)."

4. Page 3, line 32: Ovadnevaite et al study better characterised as a coastal rather than oceanic location. Furthermore, the aerosol size distribution and chemical composition they studied is not representative of marine aerosol at large – I suggest removing "with potential global implications due to the relative significance of marine aerosol". See Heintzenberg et al (2000), for example, for comparison.

We have clarified the location and environment studied by Ovadnevaite et al. (2017) but kept "with potential global implications due to the relative significance of marine aerosol." Heintzenberg et al. (2000) include data from Mace Head in their overview of marine aerosols, and Ovadnevaite et al. (2017) specifically study North Atlantic marine air masses.

"Ovadnevaite et al. (2017) soon after showed that there may be evidence for such mechanisms in observations of primary organic aerosol (POA) from a coastal environment at Mace Head, with potential global implications due to the relative significance of marine aerosol."

5. Page 3, line 6: Around here I suggest adding the reference Lowe et al. (2016) – uses 4 surfactant models, very similar conceptually to those in this study, to model CCN spectra across a similar range of supersaturations.

The reference has been added, in the suggested and a couple of additional places. We thank the reviewer for the suggestion.

". . . and model-generated synthetic aerosol representing a variety of atmospheric environments (Lowe et al., 2016), demonstrated how surface activity and its effect on cloud condensation nuclei (CCN) activity involve complex non-linear interactions between both surface tension and bulk-to-surface partitioning in droplets."

6. Page 4, line 32: State that this is $D_{p,50}$

Done.

7. Page 4, line 34: State somewhere the numerical values of all supersaturation bands.

Done.

"The CCN counter was operated at nine different supersaturations (0.10, 0.21, 0.38, 0.51, 0.69, 0.90, 1.0, 1.3, and 1.4%) for 20 minutes each so that approximately eight complete size distributions from the DMA are sampled at each supersaturation."

8. Page 5, line 7: Is there some rationale for choosing this mass fraction? I guess to be a representative aerosol composition in a vegetation-rich region. I think it would be beneficial to add some motivating text. Furthermore, it would be have been nice to see how the results and conclusions might change subject to a varying mass fraction. I recommend including something to this effect in the revisions if possible.

We have added the following section to clarify the choice of organic–inorganic mixing ratio:

"Mixing with ammonium sulfate salt in the aqueous phase is a simple way to mimic atmospheric aging in various environments, such as cloud processing and formation of secondary inorganic aerosol in polluted air, but is also used here as a physico-chemical indicator to highlight the presence and magnitude of characteristic signatures of aqueous surface activity (Prisle et al., 2010, 2011). The specific organic–inorganic mass mixing ratio was chosen based on observations from previous work that (*i*) surface activity effects became evident in cloud droplet activation behavior of particles with more than about 50% by mass of surface active organic aerosol (Prisle et al., 2008, 2010), (*ii*) additional effects of organic–inorganic solute interactions were predicted to be most prominent for mass mixing ratios in the range of 80-95% surface active organic mass (Prisle et al., 2010, 2011), and (*iii*) among these particle compositions, the lower ratios of surface active organic are likely to be the more atmospherically relevant in general (McFiggans et al., 2006; Hallquist et al., 2009). However, as pollenkitt is a pollen grain borne POA, the actual range of organic–inorganic mixing ratios resulting from atmospheric processing remain speculative."

In the present work we have focused on surface activity effects as they vary across particle size, rather than organic–inorganic composition. One of the main reasons for this is the relative scarcity of pollenkitt sample. In order to sufficiently constrain mixing ratio variations, we would need to measure surface tensions and CCN activity across a wide range of compositions and a new sample stock solution would have been needed in each case. On the contrary, sample concentration in case of surface tension measurements and dry particle size in case of CCN activity measurements can be varied from a single stock solution. We have added the following sentence to clarify this limitation:

"A full characterization of mixing effects with the methods applied here would require preparation of fresh stock solutions for each organic–inorganic composition. Due to the relative scarcity of pollenkitt samples, measurements were therefore limited to one AS mixing ratio for each of the two pollenkitt mixtures."

9. Page 7, line 19: Is c_PK the total concentration for the bulk model and surface phase concentration in the partitioning model? If so, would be good to clarify here.

The concentration dependent surface tension is specifically parametrized as a function of solution bulk concentration, which for macroscopic (bulk) solutions for all purposes is identical to the total concentration. In sub-micron droplets, surface partitioning depletes the bulk and we correct for this when using the parametrization based on bulk concentrations to evaluate droplet surface tension. We have now tried to emphasize this in the appropriate places throughout the text, in particular Section 2.4.

10. Page 8, section 2.5: state what you are assuming for surface tension for this procedure. If sigma=sigma_w then change in Eq. 1. Otherwise the reader doesn't get this information until page 12 line 19.

In Eq. (1), we have expressed Köhler Theory in its canonical form without any assumptions. The impact of different representations of droplet surface tension, including assuming $\sigma = \sigma_w$, is explored with the different thermodynamic formulations of Köhler Theory (Prisle et al., 2008, 2010). In defining $\kappa$-Köhler Theory in Eq. (4), we have followed Petters

and Kreidenweis (2007) and stated the surface tension as that of the solution droplet–air interface. A sentence has been added to clarify that $\sigma = \sigma_w$ is assumed when inverting for $\kappa$ from $D_{p,c}$ and $SS_c$, which is the basis for considering effects of droplet surface tension via the resulting $\kappa$ values.

"Furthermore, we have assumed, for the purposes of this calculation, that $\sigma = \sigma_w$ such that any effects of changes in droplet surface tension on cloud droplet activation are captured by the evaluated effective $\kappa$ parameter."

11. Page 8, line 12: Is this Dp,c the same as Dp,50? C.f. previous comments regarding activation and critical sizes.

Yes, it has been changed to $D_{p,50}$.

12. Page 19, line 15: Do you mean Ragweed rather than poplar? The solid green curve for poplar is > k_AS for all sizes.

Yes, the reviewer is correct.

"For poplar, this effect is predicted to be sufficiently pronounced to make even the intrinsic hygroscopicity of the pure pollenkitt particles larger than ammonium sulfate. For ragweed, the effect is seen for the smaller particles, which activate as more concentrated droplets."

Technical comments:

1. Cloud activation is strange terminology. I suggest changing to condensation nuclei (CN) activation, or similar, throughout

We have changed cloud activation to cloud *droplet* activation, which is the more precise term, throughout the manuscript.

2. Page 5, line 26: remove "(increasing d)"

Done.

3. Page 8, Eq. 4: Replace M_w and rho_w with partial molar volume as in Eq.1. Then no need to redefine. R also already defined under Eq.1

We have written $\kappa$-Köhler Theory as originally formulated by Petters and Kreidenweis (2007). Implicit in this expression of Köhler Theory is the assumption that the droplets being considered are dilute enough such that $\nu_w = M_w/\rho_w$. This is a common assumption since $\nu_w$ is often not a known as a function of concentration, but not the general expressions of Eqs. 1 and 4.

4. Page 9, line 11: section 3.2 ?

Yes, it has been corrected.

5. Page 11, line 5: change weight –> mass

Done. It has also been changed in Tables 2 and 3.

6. Page 12, line 20: Define growth factor

Done.

"The relative diameter growth factor at the critical point of activation –the ratio of droplet size at activation to the dry particle size, here as $d_c/D_{p,50}$– decreases with particle size ..."

7. Page 12, line 25: "high droplet total concentration", a bit confusing replace with "high total pollenkitt concentrations" or similar (if that's what you mean).

We have attempted to clarify the sentence:

"This effect is reducing $\kappa$ values to a greater extent than any simultaneous increase in effective $\kappa$ from reduced surface tension, even at the highest total solute concentrations in the droplets."

8. Page 12, line 28: nether -> neither

Oops, thank you.

9. Figure 3: explain box whiskers in caption

An explanation has been appended to the figure caption:

"Error bars are standard deviations from the $D_{p,50}$ calculated from the SMCA data inversion and standard deviations of the $\kappa$ values inverted from pairs of $D_{p,50}$ and $SS_c$."

[revised manuscript text omitted]

often not met by our measurements, as seen in Section 3. Furthermore, we have assumed, for the purposes of this calculation, that $\sigma = \sigma_w$ such that any effects of changes in droplet surface tension on cloud droplet activation are captured by the evaluated effective $\kappa$ parameter. For measurements of each pollenkitt and pollenkitt-ammonium sulfate mixture, averages and standard deviations in $\kappa$ are calculated for each supersaturation and across all supersaturations.

$$\kappa = \sum_i \epsilon_i \kappa_i,$$

~~where $\epsilon_i$ and $\kappa_i$ are the volume fraction and $\kappa$ of the $i^{th}$ component, respectively. This relation is used together with the known $\kappa$ value of pure AS to derive effective $\kappa$ values pertaining specifically to the pollenkitt fraction of mixed particles. These values are then compared with the corresponding $\kappa$ values derived from pure pollenkitt particles, allowing us to evaluate the impact of pollenkitt–ammonium sulfate aqueous phase interactions in the activating droplets.~~

**3 Results**

**3.1 Pollenkitt CCN activity**

Measured CCN activity, given as particle critical dry diameter ($D_{p,50}$) vs. supersaturation ($SS$), for the six different pollenkitts and the two mixtures with ammonium sulfate is shown in  Fig. 1. All the pure pollenkitts exhibit similar CCN activity, with some moderate variations between different species. Overall, the CCN activity of the six pollenkitts increase in the order ragweed $\approx$ sunflower $<$ olive $<$ dandelion $<$ Kentucky bluegrass $<$ poplar at a supersaturation of 1.0%, and ragweed $<$

sunflower $<$ olive $<$ dandelion $<$ poplar $<$ Kentucky bluegrass at a supersaturation of 0.2%. See panel (a) in  Fig. 1. Hence, the order of increasing CCN activity varies only little with  supersaturation and thus particle size.

5     For each particle composition, linear fits have been made to the plots of $\ln D_{p,50}$ vs. $\ln SS$. According to standard equilibrium Köhler theory, the slope of these lines should ideally be $-3/2$, and any deviations from this value would indicate the presence of size-dependent variation in particle CCN activation (albeit not which underlying size-dependent property is responsible for the variation). In cases such as this, where the well-constrained laboratory generated particle composition can be assumed to be the same for particles of all sizes generated from the same stock solution, a size-dependent variation in CCN

10 activity could result from either aqueous solubility effects in the  droplet bulk or from surface tension effects  pertaining to the droplet surface.

    Fitted slope values with standard deviations and goodness of fit, $\chi^2 = \sum_i (O_i - E_i)^2$ are given in Table 2. The goodness of fit is the sum of differences between observed outcomes $O_i$ and expected outcomes $E_i$. For dandelion, Kentucky bluegrass, and sunflower, we see no significant deviation from a slope of $-3/2$, and thus no immediate indication in our data of size-dependent variations in CCN activity of these pure pollenkitts. For olive, poplar, and ragweed, the slopes deviate from $-3/2$ beyond the standard deviation of the fit. If size-dependent CCN activity effects are introduced by pollenkitt surface activity, we would therefore expect to find them specifically for these pollenkitts among our samples. Of these, the deviation is strongest

5 for poplar, which is also the most CCN active pollenkitt over most of the particle size range studied. Ragweed is the least CCN active of the six pollenkitts, and has a slope similar to that of olive. We therefore chose to study possible effects of surface tension closer for the cases of ragweed and poplar pollenkitts (see Section  3.2).

    For pollenkitt and ammonium sulfate mixtures (panel (b) in  Fig. 1),  AS enhances CCN activity of ragweed, as might immediately be expected upon addition of a highly hygroscopic salt. However, for poplar, the enhancement

10 of CCN activity is only seen for larger particles, whereas CCN activation is suppressed for smaller particles. In case of both mixtures, the slope of the $\ln D_{p,50}$ vs. $\ln SS$ plots change significantly compared to the respective pure pollenkitt particles. This indicates the presence of now significant size-dependent effects on CCN activity introduced by the interaction between pollenkitt and the inorganic salt via either solubility or surface tension effects.

**3.2   Pollenkitt surface activity**

**Table 3.** Pollenkitt surface activity: Szyszkowski fitting parameters according to  Eq. (3) for bulk aqueous solutions of ragweed and poplar pollenkitts and their mixtures with ammonium sulfate.

[revised manuscript text omitted]

10 ~~We finally derived hygroscopicity values $\kappa$ from experimental CCN activity data, specifically for poplar and ragweed pollenkitts in particles mixed with ammonium sulfate, assuming the ZSR volume additivity of individual components on the overall particle hygroscopicity . In Figure **??**, these $\kappa_{PK}^{\text{eff,mix}}$ values are shown for poplar and ragweed together with pure pollenkitt $\kappa$ values. For ragweed , interactions with ammonium sulfate clearly increase pollenkitt effective hygroscopicity.For poplar, pollenkitt effective hygroscopicity in mixed particle is lower for smaller particles and somewhat higher for larger particles , compared to similar sized particles of pure pollenkitt . As the stronger surfactant, in mixed particles, poplar pollenkitt~~

15

**4 Modeling CCN activity of pollen grains**

Pollenkitt is studied here as an example of atmospheric biogenic primary organic aerosol. By resolving the specific water interactions in terms of cloud forming potential of pollenkitt, we are able to decouple the contributions of bare (defatted) pollen grains or their fragments, soluble and surface active pollenkitt, and potential inorganic components such as ammonium

20 sulfate formed by atmospheric processing of pollen. With the intrinsic hygroscopicity of each of these components constrained, the overall response of pollen grains can be modeled for a wide variety of conditions without the specific knowledge from immediately corresponding measurements.

[Figure]

**Figure 8.** (a) Modeled CCN activity in terms of dry particle diameter ($D_p$) vs. critical supersaturation ($SS_c$) for whole ragweed pollen grains with different contributions from pollenkitt and ammonium sulfate. CCN activity derived from hygroscopicity values for pure ammonium sulfate and a range of valued measured for different whole pollen grains by Griffiths et al. (2012), and for SSP measured by Steiner et al. (2015), are shown for comparison; (b) Modeled Köhler curves for a 15 $\mu$m ragweed pollen grain with different contributions from pollenkitt (15% mass fraction) and ammonium sulfate (in 1:4 mass ratio relative to pollenkitt).

As an example, we calculate the critical supersaturations for ragweed pollen grains of different sizes, first assumed to be composed of a spherical, insoluble core with density of $\rho_{\text{insol.}} = 1.28$ g cm$^{-3}$, corresponding to the value measured at 100% RH (Harrington Jr. and Metzger, 1963), and a pollenkitt coating with the mass fraction of $W_{p,\text{PK}} = 0.15$ as measured by Lin et al. (2013) and density of $\rho_{\text{PK}} = 1.2$ g cm$^{-3}$ as used in this work. Assuming volume additivity of the insoluble core and pollenkitt coating according to Eq. 2, the volume fraction of pollenkitt $\epsilon_{\text{PK}}$ in the pollen grain is related to its mass fraction by

$$\epsilon_{\text{PK}} = W_{p,\text{PK}} \left( W_{p,\text{PK}} + W_{p,\text{insol.}} \frac{\rho_{\text{PK}}}{\rho_{\text{insol.}}} \right)^{-1} \tag{5}$$

where the insoluble core and pollenkitt mass and volume fractions are related as $W_{p,\text{insol.}} = 1 - W_{p,\text{PK}}$ and $\epsilon_{\text{insol.}} = 1 - \epsilon_{\text{PK}}$. From this, an overall effective $\kappa_{\text{pollen}}$ for the ragweed pollen grain can be calculated from

$$\kappa = \sum_i \epsilon_i \kappa_i \tag{6}$$

and assuming individual hygroscopicity values of $\kappa_{insol} = 0$ for the insoluble core and $\kappa_{PK} = 0.14$ for ragweed pollenkitt as derived from our measurements here. Critical supersaturations $SS_c$ are then calculated from $\kappa_{pollen}$ as a function of pollen grain diameter $D_p$ according to Eq. 10 from Petters and Kreidenweis (2007):

$$SS_c = \left( \exp \left[ \sqrt{\frac{4A^3}{27D_p^3 \kappa}} \right] - 1 \right) \times 100\% \tag{7}$$

where

$$A = \frac{4\sigma M_w}{RT\rho_w} \tag{8}$$

With a fixed mass fraction of pollenkitt, the volume fractions of each component are also constant according to Eq. 5. The overall ragweed pollen grain hygroscopicity $\kappa_{pollen}$ is therefore not a function of pollen grain diameter under these assumptions.

Following a similar procedure, we then add ammonium sulfate to the ragweed pollen grains, assuming that the relative mass ratio of $0.15 : 0.85$ between pollenkitt and the insoluble core is maintained, and that the mass ratio between ammonium sulfate and pollenkitt is $1 : 4$, as in this work. This yields the relations $W_{p,insol} = \frac{0.85}{0.15} W_{p,PK}$ and $W_{p,PK} = 4W_{p,AS}$ between the mass fractions of individual pollen components, from which each mass fraction can be determined under the constraint that $W_{p,insol} + W_{p,PK} + W_{p,AS} = 1$. As before, with the assumption of volume additivity, these mass fractions yield invariant volume fractions of insoluble core, pollenkitt, and ammonium sulfate. The combined volume fraction of soluble pollenkitt–ammonium sulfate mixture, $\epsilon_{PK-AS} = \epsilon_{PK} + \epsilon_{AS}$, is then determined from Eq. 5 by substituting $\epsilon_{PK-AS}$ for $\epsilon_{PK}$, the combined pollenkitt–ammonium sulfate mass fraction $W_{p,PK-AS} = W_{p,PK} + W_{p,AS}$ for $W_{p,PK}$, and the pollenkitt–ammonium sulfate mixture density $\rho_{PK-AS}$, as given from Eq. 2 with $\rho_{AS} = 1.769$ g cm$^{-3}$, for $\rho_{PK}$. With the volume fractions $\epsilon_{PK-AS}$ and $\epsilon_{insol} = 1 - \epsilon_{
[revised manuscript text omitted]

---

## Author Response (AR2)

**Response to Reviewer's comments and suggested minor revisions.**

We are happy that the reviewer now finds the clarity of the manuscript much improved and once again thank for the many useful suggestions.

The aim of this work is to study effects of surface activity in CCN activation of real atmospheric complex organic aerosol. While the analysis of CCN activity in the framework of Köhler theory lends itself readily to both study of surfactant effects and characterization of small amounts of sample as described, the characterization of aerosol-water interactions which govern cloud activation and hygroscopic water uptake has implication and applications for a wide range of phenomena beyond cloud microphysics, which are governed by the same underlying water interactions.

We therefore emphasize both previous work that has motivated the present study and specifically our choice of pollenkitt as POA model system, as well as highlight some of the most prominent and immediate potential implications and applications of the present results, as suggestions for further investigation. In response to earlier reviewer comments, we specifically illustrate how our present results enable a significant generalization of hygroscopic interactions of pollen and pollen fragments with water in the atmosphere and present a detailed scheme outlining how this could be immediately implemented in atmospheric chemistry and transport models. We here note that Figure 8 covers predictions for particle diameters between 30 nm and 100 micron, including size ranges relevant for both pollen grains and SSP. The Köhler curve in the right panel of Figure 8 is shown as an example of water uptake over a range of ambient humidities for the specific size of 15 micron, the mean diameter observed in one study for ragweed pollen grains, but similar curves can be obtained for all sizes across the spectrum in the left panel, and for any given humidity range. We explicitly illustrate how to include inorganic components formed by atmospheric processing, for the fraction specifically constrained by out measurements, but also how variable amounts of pollenkitt and inorganic components may be included across particle sizes. These are some of the specific examples given of the ranges of atmospheric conditions which may be represented using the present results.

By water interactions, we therefore do mean the full breadth of potential interactions between complex OA, including pollenkitt as part of pollen and their fragments, which can be inferred from the CCN and Köhler analysis of the present work. Among these, as mentioned, are properties related to atmospheric transport, ageing, and cloud formation from pollen as giant CCN, IN, and SSP, and biological functions related to modulation of pollen wettability and adhesion. However, we also emphasize that our results highlight effects of surface activity in the general context of complex OA, and are not only relevant for POA or pollen systems. A full account of all the potential water interaction properties which may be inferred from the present work and their uses, is beyond the scope of this work, but as mentioned, some are the topics of ongoing work.

We have now further clarified these points throughout the manuscript, e.g. in the Abstract, Introduction and Sections 3.4, 4, and 5.

[revised manuscript text omitted]